# A novel hydrographic gridded data set for the Northern Antarctic Peninsula

Tiago S. Dotto[1,a], Mauricio M. Mata[1,2], Rodrigo Kerr[1,2], Carlos A. E. Garcia[1,2]

[1]Laboratório de Estudos dos Oceanos e Clima, Instituto de Oceanografia, Universidade Federal do Rio Grande-FURG, Brazil
5 [2]Programa de Pós-Graduação em Oceanologia, Instituto de Oceanografia, Universidade Federal do Rio Grande-FURG, Brazil
[a]Now at Centre for Ocean and Atmospheric Sciences, School of Environmental Sciences, University of East Anglia, UK

*Correspondence to*: Tiago S. Dotto (t.segabinazzi-dotto@uea.ac.uk), Mauricio M. Mata (mauricio.mata@furg.br), Rodrigo Kerr (rodrigokerr@furg.br), Carlos A. E. Garcia (dfsgar@furg.br)

**Abstract.** The Northern Antarctic Peninsula (NAP) is a highly dynamic transitional zone between the subpolar-polar and 10 oceanic-coastal environments, and it is located in an area affected by intense climate change, including intensification and spatial shifts of the westerlies as well as atmospheric and oceanic warming. In the NAP area, the water masses originate mainly from the Bellingshausen and Weddell Seas, which create a marked regional dichotomy thermohaline characteristic. Although the NAP area has relatively easy access when compared to other Southern Ocean environments, our understanding of the water masses distribution and the dynamical processes affecting the variability of the region is still limited. That limitation is closely 15 linked to the sparse data coverage, as commonly is the case in most of the Southern Ocean environments. This work provides a novel seasonal three-dimensional high-resolution hydrographic gridded data set for the NAP (version 1), namely the NAPv1.0. Hydrographic measurements from 1990 to 2019 comprehending data collected by Conductivity, Temperature, Depth (CTD) casts, sensors of the consortium Marine Mammals Exploring the Oceans Pole to Pole (MEOP) and Argo floats have been optimally interpolated to produce maps of in situ temperature, practical salinity and dissolved oxygen at ~10 km 20 spatial resolution and 90 depth levels. The water masses and oceanographic features in this regional gridded product are more accurate than other climatologies and state estimate products currently available. The data sets are available in netCDF format at https://www.goal.furg.br/producao-cientifica/supplements/203-goal-gridded-nap and at https://doi.org/10.5281/zenodo.4420006 (Dotto et al., 2021). The novel and comprehensive data sets presented here for NAPv1.0 product are a valuable tool to be used in studies addressing climatological changes in the unique NAP region since 25 they provide accurate initial conditions for ocean models and improve the end of the 20th and early 21st-century ocean mean-state representation for that area.

## 1 Introduction

The Northern Antarctic Peninsula (NAP; Kerr et al., 2018a) encompasses the Bransfield and Gerlache Straits, the northwestern Weddell Sea, the southernmost Drake Passage and the northern end of the West Antarctic Peninsula 30 environments (Fig. 1a). The NAP is a sensitive region to climate changes because it is located under the influence of the

westerly winds, which are prone to current intensification and poleward migration (Marshall, 2003; Swart and Fyfe, 2012; Lin et al., 2018). The region also receives considerable freshwater input from the melting of glaciers from the Antarctic Peninsula (Cook et al., 2016; Rignot et al., 2019). Moreover, the NAP has been showing significant alterations in its water masses physico-chemical properties, such as warming, freshening, and acidification (Gordon et al., 2000; Meredith and King, 2005;

Hellmer et al., 2011; Dotto et al., 2016; Kerr et al., 2018b; Lencina Avila et al., 2018; Ruiz Barlett et al., 2018), a decline in sea ice extension and shortening of sea ice cover season (Stammerjohn et al., 2008; Turner et al., 2013), as well as in its marine ecosystem (Moline et al., 2004; Montes-Hugo et al., 2009; Mendes et al., 2013).

Water masses with different properties and origins reach the NAP and mix to form the dense water masses that sink and fill the deep basins of the Bransfield Strait (Gordon et al., 2000; Dotto et al., 2016; Huneke et al., 2016; van Caspel et al.,

2018). High Salinity Shelf Water (HSSW) and Low Salinity Shelf Water (LSSW) from the Weddell Sea enter the NAP from the east, mainly contouring the Antarctic Peninsula (von Gyldenfeldt et al. 2002; Heywood et al., 2004; Thompson et al., 2009; Collares et al., 2018), and then sinking along the slope and at the Bransfield Strait's several canyons (van Caspel et al., 2018; Fig. 1b). A branch carrying modified HSSW flows southwestward along the continental shelf to the west of the Antarctic Peninsula and reaches as far as the Gerlache Strait (Sangrà et al. 2017; Kerr et al., 2018b). Conversely, modified Circumpolar

Deep Water (mCDW) from the Bellingshausen Sea – a relatively warm, salty, nutrient-rich and deoxygenated water mass derived from the intermediate waters of the Antarctic Circumpolar Current (ACC) and modified over the western Antarctic Peninsula continental shelf – enters the NAP mainly through the Bransfield Strait western basin (Smith et al., 1999; Ruiz Barlett et al., 2018), flowing northeastward along the South Shetland Islands slope as the Bransfield Current (Sangrà et al., 2011; Fig. 1b). The mCDW also intrudes into the Bransfield Strait from the southern Drake Passage between King George and

Elephant Islands (López et al., 1999; Gordon et al., 2000). Therefore, a cyclonic circulation pattern is created within the Bransfield Strait from currents of completely different origins defining a strong transitional signature for that whole area (García et al., 2002). Two main fronts are observed in this region: the Bransfield Front and the Peninsula Front (Sangrà et al. 2011; Fig. 1b). The Bransfield Front is the mid-depth front separating the warm waters flowing along with the Bransfield Current and the cold waters within the deep basins of the Bransfield Strait. The Peninsula Front separates, in shallow depths,

the cold waters from Weddell influence and the Bransfield Strait waters. In the Gerlache Strait, mCDW from the Bellingshausen Sea also intrudes from the south and through the gaps near the Anvers Island (Smith et al., 1999; García et al., 2002; Torres Parra et al., 2020; Fig. 1b). The mCDW interacts with the HSSW-sourced waters, and a relatively colder mixture leaves the Gerlache Strait towards the western basin of the Bransfield Strait (Niiler et al., 1991; Zhou et al., 2002; Savidge and Amft, 2009). In the northwestern Weddell Sea, the Antarctic Slope Front separates the coastal from the open ocean waters

circulating within the Weddell Gyre (Heywood et al., 2004). Part of the dense waters that circulate in the Powell Basin eventually leak through the Philip Passage and are exported from the Weddell Sea as Antarctic Bottom Water (Franco et al., 2007; Fig. 1b). Other routes of Antarctic Bottom Water export from the Weddell Sea are the Orkney, Bruce, and Discovery Passages located to the east of South Orkney Island (Naveira Garabato et al., 2002).

For the past fifty years at least, the deep waters of the Bransfield Strait have shown significant trends of freshening
and lightening, with impacts on the volume of these regional dense waters (Azaneu et al., 2013; Dotto et al., 2016; Ruiz Barlett
et al., 2018). Considering that these waters are formed by a parcel of ~60-80% of HSSW+LSSW (Gordon et al., 2000; Dotto
et al., 2016), the freshening signal may be driven by modification of the water masses sourced in the Weddell Sea continental
shelf (van Caspel et al., 2015, 2018), possibly due to the melting of ice shelves and glaciers in the Antarctic Peninsula (Cook
et al., 2016; Rignot et al., 2019). In addition, the decreasing of the sea ice concentration and the shortening of the sea ice season
in the NAP (Stammerjohn et al., 2008; Turner et al., 2013) may also play a role in the freshening trend due to a reduction of
the salt flux into these shelf waters (Hellmer et al., 2011).

Such mixture of distinct water masses creates good conditions for the development of a rich biological activity in the
NAP. High concentrations of phytoplankton have been documented in the Bransfield and Gerlache Straits, with implications
for local carbon dioxide uptake (Costa et al., 2020; Monteiro et al., 2020a, b). The high primary production also supports an
abundant krill stock (Atkinson et al., 2020), which is the base of the food chain up to top predators (Seyboth et al., 2018). If
the changes reported for the region continue (e.g., sea ice cover and seasonality declining, land ice melting, atmospheric and
oceanic temperature warming), it is expected an imbalance or even a collapse in the regional food chain (Ferreira et al., 2020).
A change in the composition of phytoplankton, from diatoms to cryptophytes, has already been observed in the NAP (Mendes
et al., 2013), as well as a reduction in the recruitment of juvenile krill and a general decline of krill stocks in areas of the
Atlantic sector of the Southern Ocean, including the NAP (Atkinson et al., 2020).

A better understanding of the NAP oceanic and shelf circulation, as well as the connection to the Weddell and
Bellingshausen Seas, is important to help to assess the evolution and impacts of the ocean on the ice shelves and glaciers
melting downstream (Cook et al., 2016; Rignot et al., 2019). Currently, the overall change in the glaciers' area is relatively
small in the Bransfield and Gerlache Straits compared to the adjacent Bellingshausen coastline, which is under the influence
of warmer waters. However, changes in proportions of the inflowing water masses could potentially lead these regions to a
Bellingshausen Sea-like scenario, where warmer oceanic conditions shape the local glacier fronts behaviour and higher melting
rates are observed (Cook et al., 2016). Ruiz Barlett et al. (2018) suggest that persistent northerly and westerly wind conditions,
concurrent with La Niña and positive phases of the Southern Annular Mode (SAM), are favourable drivers for mCDW inflow
into the NAP. These warm inflows are likely facilitated by the poleward displacements of the oceanic fronts associated with
the ACC during La Niña events (Loeb et al., 2009). Additionally, positive SAM may restrict the connections between the
Weddell Sea and the Bransfield Strait (Renner et al., 2012), with the potential of reducing the inflow of cold waters into the
NAP (Dotto et al., 2016). The ongoing changes in the Southern Hemisphere wind pattern, i.e. poleward and intensification
associated with positive trends in the SAM, and an increase in the energetics of the ACC (Meredith and Hogg, 2006; Hogg et
al., 2015) could then lead to higher mCDW inflow into the NAP by advection (e.g., Ruiz Barlett et al., 2018) and/or eddies
shed by the ACC (Martinson and McKee, 2012; Couto et al., 2017). Although the strengthening of winds tends to increase the
inflow of warm waters toward the west Antarctic Peninsula continental shelf, it does not necessarily mean higher ice shelf
basal melting (except for the shallower ice shelves). Conversely, models suggest a reduction of sea ice concentration associated

with an enhancement of the upper ocean heat fluxes (Dinniman et al., 2012). In the NAP, mixing rate measurements are scarce, hampering the understanding of how heat fluxes in the upper and deeper ocean will respond to the future wind changes in the region. In addition, there is still a lack of information regarding the water masses fluxes and mixing between the Powell Basin in the Weddell Sea and the Bransfield Strait (Thompson et al., 2009; Azaneu et al., 2017), which would help to increase our understanding of the complex coastal Antarctic system.

In this context, forecasts of the evolution of the dynamics of these regional seas and their interconnections in future climate change scenarios are needed. In general, in situ measurements in the Southern Ocean are scarce in space and time due to logistical constraints. Moreover, that scarceness impacts a proper understanding of the oceanographic processes and their interactions with the atmosphere and cryosphere, which requires robust and long-term Southern Ocean observing systems to provide high-quality measurements of the different ocean and cryosphere compartments (Meredith et al., 2013; Newman et al., 2019). Hence, high-resolution regional models are needed to integrate these data and to simulate future scenarios. In this sense, quality-controlled gridded products to feed these models are important to properly create the initial conditions, in terms of the 3D structure of the water masses, which are needed to capture the main oceanographic and glaciological interactions and, consequently, their future evolution.

Here, we present the version 1 of a hydrographic gridded product, called NAPv1.0, comprising robust high-resolution measurements from different platforms: ship-based, Argo profilers and tagged marine mammals. The gridded product covers the period of 1990 to 2019, and it has in situ temperature, salinity and dissolved oxygen fields. NAPv1.0 can be used for several applications, including input data for ocean and climate models initialization/assessment and ocean reanalysis evaluation, as well as to produce and reconstruct biogeochemical properties. Finally, the NAPv1.0 represents the ocean mean-state on seasonal scales for the NAP in the end 20th-century and early 21st-century.

## 2 Hydrographic data and ancillary data set

In situ temperature, practical salinity and dissolved oxygen (DO) data from five main sources were used to build the data base for the gridded product. Conductivity-temperature-depth (CTD) profiles were acquired from the World Ocean Database (WOD; https://www.ncei.noaa.gov/products/world-ocean-database) and Pangaea (https://www.pangaea.de/) for the period of 1990 to 2019. In addition, CTD profiles from the Brazilian High Latitude Oceanography Group (GOAL; Mata et al., 2018) are included for the years spanning from 2003 to 2019 while data from Hutchinson et al. (2020; https://doi.org/10.5281/zenodo.3357972) are included for the region off Larsen C Ice Shelf. Since 2003, the GOAL has been conducting research cruises on a quasi-annual basis in the NAP area during summer periods. Most of the efforts have focused on the Bransfield and Gerlache Straits, due to their relatively easier access, logistics and favourable meteo-oceanographic conditions. Argo floats data were retrieved from Coriolis website (http://www.coriolis.eu.org/). Profiles from tagged-marine mammals were obtained from the consortium Marine Mammals Exploring the Oceans Pole to Pole (MEOP; http://www.meop.net/). Although all platforms measure in situ temperature and conductivity data, only CTD and Argo provide

DO. The study region was limited to 42°-68°W and 59.5°-66°S, and the time span was from 1990 to 2019 (Figure 2). The data sets were grouped in different seasons to allow a better representation of the region: summer (January to March), autumn (April to June), winter (July to September) and spring (October to December). Only reliable data according to the quality flags of each data set were used. In addition, a visual inspection was carried out to double check any spurious data. Once the profiles were quality controlled, each profile was linearly interpolated onto 90 standard depths ranging from 5 m to 6000 m depth

(Table 1). The depth spacing ranges from 5 m in the upper 50 m to 500 m below 4000 m depth in oceanic regions. The spacing between layers were arbitrarily selected.

A total of 37009 profiles of in situ temperature and 32562 of salinity were used in this study (Fig. 3a,b). Most of these data were sampled by MEOP, followed by CTD and Argo. In autumn and winter, the amount of data collected by MEOP can be as high as 18 times compared to the other platforms. Summer and autumn were the seasons with the higher amount of data

collected, followed by winter and spring. Note, however, that the high amount of data in autumn and winter is due to MEOP measurements (Fig. 3). In regards of DO, a total of 1810 profiles were used (Fig. 3c). Most of the data was sampled in summer by ship-based CTD. From this total, 701 DO profiles were acquired from GOAL in the Bransfield and Gerlache Straits, eastern Bellingshausen Sea and northwestern Weddell Sea. All years, except 2012, were covered by CTD temperature and salinity samples (Fig. S1a,b). Argo and MEOP data were available from 2002 and 2008, respectively, for the study region (Fig. S1).

DO had a better sampling covering after 2003, when the GOAL carried out studies in the region quasi-annually (Fig. S1c).

Most of the samples were collected in the upper ocean (up to ~2000 m depth; Fig. 3d-f). This happens because Argo profiles have a limit depth range up to 2000 m and the MEOP data are mostly restricted to the upper ~1000 m due to the physiology and behaviour of the animals (Fig. S2). Below 2000 m depth, our climatology relies only on CTD data. Consequently, seasons with reduced CTD profiles will have less representativeness. In addition, the study region is dominated

by shallow seas (Fig. 1a), and deep areas are found only adjacent to the Antarctic Peninsula. The first standard depth (i.e., 5 m) had less samples than the second standard depth (i.e., 10 m). For this reason, we repeated the second level in the gridded product as our first level.

The different platforms have different accuracies given their distinct frequency sampling. The initial CTD accuracy reported for the World Ocean Database typically ranges from ±0.001°-0.005°C for temperature and approximately ±0.003-

0.02 for salinity (Boyer et al., 2018). For the Pangaea data set, the accuracy is documented as ±0.001°C for temperature and approximately ±0.003 for salinity (Driemel et al., 2017). The accuracy of the GOAL CTD data is also ±0.001°C and ±0.003 for temperature and salinity, respectively. For the DO, the accuracy can be 2% of saturation of more, depending the sensor (Boyer et al., 2018). The GOAL DO data showed a mean absolute difference of ~0.09 mL L$^{-1}$ between the double sensors used in the cruises. The accuracy of the delayed-mode Argo is ±0.002°C and ±0.01 for temperature and salinity, respectively

(Wong et al., 2020). The initial accuracy of the Argo DO sensors range from 2-5% of saturation, according to the manufacturers. The CTD-satellite relay data logger (CTD-SRDL) from MEOP is reported to have an accuracy of ±0.02-0.04°C for temperature and ±0.03 for salinity (Treasure et al., 2017; Siegelman et al., 2019).

Given the higher resolution and accuracy, the CTD data was used for a reasonableness check to verify if the gridded products can represent the expected and accepted values of the hydrographic properties in the NAP. For this exercise, we chose

to compare only those CTD profiles closer than ~6 km from each grid points of the climatologies evaluated. In addition, we present a section comparing different climatologies for the NAP. Summer-averaged temperature and salinity from the World Ocean Atlas 2018 (WOA; https://www.ncei.noaa.gov/access/world-ocean-atlas-2018/; Locarnini et al., 2019; Zweng et al., 2019) was used for comparison with the NAPv1.0 results. WOA has a spatial resolution of 1/4˚ and 102 vertical levels. DO is available only in coarser resolution, thus it is not used here. The Commonwealth Scientific and Industrial Research

Organisation (CSIRO) Atlas of Regional Seas (CARS; http://www.marine.csiro.au/~dunn/cars2009/; Ridgway et al., 2002) was also used in the same way. CARS has a spatial resolution of 1/2˚, 79 vertical levels and it comprises gridded fields of the mean ocean temperature, salinity and DO. The summer season mean conditions are estimated from the annual and semi-annual coefficients distributed with the climatological fields; however, these coefficients are limited to the mid- and upper-levels. Finally, the Southern Ocean State Estimate (SOSE; http://sose.ucsd.edu/; Mazloff et al., 2010), coupled with biogeochemical-

sea ice-ocean state estimate (Verdy and Mazloff, 2017), was used to compare with our regional gridded product. Januaries to Marchs outputs of the runs called iterations 106 and 133 were averaged to produce a summer-mean state for the period 2008-2018 (the biogeochemical outputs are not available prior to this period). Potential temperature, salinity and DO were selected with a spatial resolution of 1/6˚ and 52 vertical levels. Conservative temperature ($\Theta$), absolute salinity ($S_A$) and neutral density ($\gamma^n$) were then computed for all data sets prior to the assessment (Jackett and McDougall, 1997; McDougall and Barker, 2011).


### 3 Methods

The scattered hydrographic data were objectively interpolated onto a regular grid of ~10 km resolution. The grid spacing is 0.09˚ along latitudes and 0.2˚ along longitudes (i.e., 0.09˚ latitude × 0.09˚/cos(63˚S) longitude, where 63˚S is the mean latitude of our domain). The limits of the domain are 59.5˚S to 66˚S and 42˚W to 68˚W. The study region covers the

transitional oceanographic regimes between the adjacent areas and the NAP, hence representing in more details the regional oceanic features. We note that even with the addition of distinct platforms, the area of the western Weddell Sea is still poorly sampled (Fig. 2). Due to this lack of data coverage, we masked that region from the climatology. Bathymetry and land masks were applied using the 1 arc-minute ETOPO1 Ice Surface data set (https://www.ngdc.noaa.gov/mgg/global/) interpolated onto a grid of same resolution.

The objective interpolation uses scattered observations to generate a regular gridded and smoothed version of the original data field analysed (Bretherton et al., 1976; Thomson and Emery, 2014). The method is based on the Gauss-Markov theorem, which consists of an application of linear estimation techniques for interpolating data from the multiple platforms combined. While this improves the spatial coverage, it also smooths out the temporal and spatial variability, resulting in a pattern that is more representative of the large-scale mean state rather than of a specific period. The interpolation of each grid

point is affected by the neighbouring data that fall within the smoothing lengthscales (or radius of influence) and the relative error chosen a priori. The interpolation method also assumes a weight to sum all observations within a specific smoothing

lengthscale. The weight is a function of the distance to the grid point and the location of the observation only. Because the objective interpolation fits a Gaussian function, the nearest neighbours have higher weight than the data located closer to the edge of the radius of influence. A series of tests were made to find the appropriate smoothing lengthscale and the a priori relative error in order to find a balance between smoothness and feature representativeness. The final smoothing lengthscale (i.e. the radius of influence of the interpolation) chosen was 1° in latitude and longitude, and the a priori relative error allowed was set to 0.2 for the objective interpolation algorithm. The same constants were set for all depth levels, all variables (i.e., in situ temperature, practical salinity and DO) and all seasons. Regions where the mapping relative error was higher than 0.5 were excluded. The matrices of relative errors are provided together with the gridded variables, so the user can choose the relative error level to work with (between 0 and 0.5). The representativeness of the interpolated maps decreases as the number of measurements being used reduces. This is reflected in higher relative errors, such as seen closer to the (i) Weddell Sea, (ii) regions where the gaps between data are large, and (iii) at deeper levels (Fig. 4). The reduction of the amount of data among the seasons also affects the interpolation, mainly in the deeper levels where less CTD profiles are available (Fig. 4). In regions of the upper ocean where a vast number of MEOP profiles are available (i.e., Bransfield Strait, Gerlache Strait and WAP), the errors are comparable among the seasons (Fig. 4). For the purposes of this study, $\Theta$, $S_A$ and $\gamma^n$ are presented.

## 4 Results and discussion

### 4.1 Reasonableness check of the gridded product

The term reasonableness check is used here to ensure that the gridded data set meets the expected range, type or value based on its individual measurements. We want to examine how reasonable the NAPv1.0 can represent the CTD profiles closer than ~6 km of each grid point. Although the data sets are not independent, this comparison shows how well the interpolation captured the real data set's magnitude, structure and spatial variability.

The NAPv1.0 shows a good agreement with the CTD profiles for the hydrographic parameters. Approximately 62%, 70%, 74% and 65% of the data points fall within the difference range of ±0.2°C for summer, autumn, winter and spring, respectively (Fig. 5a). The higher percentage of agreement in autumn and winter is due to the reduced number of CTD profiles in those seasons, which become more influential in the interpolation in those cases. The higher $\Theta$ difference, as expected, occurs in the upper ocean due to interannual variability (Mendes et al., 2013; Gonçalves Araújo et al., 2015) that is smoothed out in the interpolation process (Fig. S3a,b).

A good agreement between the NAPv1.0 and the CTD profiles is also observed for $S_A$. Approximately 67%, 74%, 84% and 75% of the data points fall within the difference range of ±0.04 g kg$^{-1}$ for summer, autumn, winter and spring, respectively (Fig. 5b). The $S_A$ differences over the water column are also higher in the upper levels due to the interannual variability, and smaller towards the deep ocean (Fig. S3c,d), where the interannual variability tends to be relatively smaller (Dotto et al., 2016; Ruiz Barlett et al., 2018).

In regards of DO, approximately 65%, 87%, 87% and 80% of the data points fall within the difference range of ±0.2 mL L$^{-1}$ for summer, autumn, winter and spring, respectively (Fig. 5c), which clearly suggests the influence of less data affecting

the resemblance between the gridded product and the CTD stations. The NAPv1.0 shows relatively higher differences related to DO data, which is a non-conservative property. The largest differences are observed in the upper ocean, but also along the water column where it is not uncommon to observe differences of up to $\pm 0.5$ mL L$^{-1}$ (Fig. S3e,f). The histograms previously presented (Figs. 3, S1 and S2) evidence the weaker representation of DO compared to the other properties in autumn, winter and spring.

## 4.2 Representation of the main hydrographic features in the NAP

The NAPv1.0 represents qualitatively well the surface thermal dichotomy regime of waters of the NAP: a warm variety with Bellingshausen Sea and ACC influence to the west, and a cold Weddell Sea-sourced water to the east (Fig. 6a). A surface thermal front splits these two regimes near the Antarctic Peninsula, likely depicting the Peninsula Front (López et al., 1999; Sangrà et al., 2011). In the surface, the coastal Weddell Sea-sourced waters are saltier and denser compared to the warm waters to the west of the NAP (Fig. 6a-c). Thus, the Peninsula Front seems to be baroclinic (Sangrà et al., 2011). The DO is relatively supersaturated everywhere, with higher concentrations found close to the James Ross Island, the Gerlache Strait and the northern end of WAP (Fig. 6d). These regions are generally associated with high primary productivity, which could explain the relatively higher values of DO (Mendes et al., 2012; Detoni et al., 2015; Costa et al., 2020).

In deeper layer, the dichotomy regime of the NAP is also still evident, reflecting the presence of warm and salty waters associated with the CDW to the west of the NAP and the Warm Deep Water (WDW; a warm, salty and poor-oxygenated intermediate water mass presented in the Weddell Sea and derived from the mixing between CDW and Winter Water) within the offshore zone of the Weddell Sea and Powell Basin (Fig. 6e,f,i,j). Intrusions of mCDW are observed in the eastern Bellingshausen Sea through deep channels, reaching the Gerlache Strait and the western basin of the Bransfield Strait. After entering the Bransfield Strait, the mCDW warm signal follows the slope of the South Shetland Islands as the narrow Bransfield Current (Niiler et al., 1991; López et al., 1999; Zhou et al., 2006; Sangrà et al., 2011). However, given the length-scales chosen, this current is not well represented in the surface maps. Although the routes of mCDW inflow are relatively well known, few studies focused on the periodicity of these inflows into the NAP (e.g., Ruiz Barlett et al., 2018) or on the fast response of these inflows associated with the wind forcing (e.g., McKee and Martinson, 2020a). The comprehension of these mechanisms is important to better resolve the heat and salt fluxes into the NAP

In the central NAP, the cold, fresh, dense and oxygen-rich Bransfield Strait characteristics are evident compared to the adjacent regions (Fig. 6e-l). The NAPv1.0 represents quite well a narrow cold and oxygen saturated path along the continental slope of the Weddell Sea at 500 m depth. At the Powell Basin, these water masses follow the bathymetry and enter the Bransfield Strait, thus connecting the Weddell Sea and the NAP (Heywood et al., 2004). These dense waters that flood the central NAP deep levels seem to follow the shortcuts of the canyons that cut across the Antarctic Peninsula continental shelf in the Bransfield Strait (Fig. 6e-h). The higher DO in that deep regions indicates a fast route of Weddell Sea shelf waters and hence a relatively small residence time (van Caspel et al., 2015, 2018). The Bransfield Strait has the highest $\gamma^n$ values compared to the surroundings (Fig. 6g,k), which is associated with the higher contribution of HSSW (Gordon et al., 2000; Dotto et al.,

2016). The water masses in the Bransfield Strait central basin are also slightly denser than in the eastern basin (Fig. 6g,k), suggesting a different origin of these waters (Gordon et al., 2000; van Caspel et al., 2018). Following $\Theta$ distribution, DO is higher on the Weddell Sea continental shelf and within the Bransfield Strait, whereas the lower values are observed associated with the older water masses mCDW and the WDW (Fig. 6h,l). WDW is another water mass that compose the deep waters of the Bransfield Strait (Gordon et al., 2000). Although the NAPv1.0 suggest the existence of an intrusion of this water mass into

the eastern basin at ~61.5°S and 55°W (Fig. 6i), we still need to increase our comprehension on the contribution to the water mass mixture and the access routes of WDW into the Bransfield Strait via Powell Basin to resolve the local circulation in more details (e.g., Thompson et al., 2009; Azaneu et al., 2017). At deeper levels, the bathymetric constraints restrict the connection of the shelf waters from the Weddell Sea, and the dense waters that sink into the Bransfield Strait remains in its quasi-pure properties (Gordon et al., 2000; Dotto et al., 2016). The horizontal maps clearly show the importance of the Bransfield Strait

in trapping the shelf waters of the Weddell Sea, making it an in situ laboratory to study temporal changes of these water masses. Given the spatial scales and the smoothing following the gridding procedure, Bransfield Strait mesoscale and submesoscale eddies are not observed in the NAPv1.0.

Seasonally, the NAPv1.0 represents a general decrease in $\Theta$, increase in $S_A$ due to sea ice formation in the region and consequently a $\gamma^n$ gain from autumn to spring in the upper layers (Figs 7-9a-d). As in summer, the Bransfield Strait is colder,

fresher, denser and oxygen-rich area compared to the adjacent regions in deeper basins (Fig. 7-9). Although slightly less visible than in summer, the narrow thermal signal along the continental slope of the Weddell Sea is observed entering the Bransfield Strait in autumn and spring. In winter, likely due to limited in situ data, this signal is not clear. Given the limitation of DO data, this field is not well represented in seasons except summer.

To exemplify the representation of the water column of NAPv1.0, we selected six sections crossing different parts of

the NAP and adjacent regions (Fig. 1a). To the southwest of the Bransfield Strait, along the section between Livingston Island and the Antarctic Peninsula, relatively warmer conditions are observed from the upper ocean to the bottom layers (Fig. 10a). An intrusion of mCDW is observed between 62.7-62.8°S, where the isotherms are sharply tilted towards the South Shetland Islands (Fig. 10a), $S_A$ is above 34.72 g kg$^{-1}$ (Fig. 10b), and the DO is relatively low < 6 mL L$^{-1}$ (Fig. 10d). The tilt of the isotherms is a signature of the Bransfield Front, which in turn is associated with the mid-depth Bransfield Current (Niiler et

al., 1991; López et al., 1999; Sangrà et al., 2011). In the section between the King George Island and the Antarctic Peninsula the warm, salty and deoxygenated mCDW is still present, however it is placed in slightly shallower levels (Fig. 10e-h). Towards the bottom, $\Theta$ decreases to the lowest values in the region, characterizing the Bransfield Strait central basin. In this section, the surface Peninsula Front is visible, associated with the cold regime from the western Weddell Sea continental shelf. In both sections, the dome-like shape of the isopycnals supports the cyclonic gyre presence in the region (Fig. 10c,g; Zhou et al., 2002;

Sangrà et al., 2011; Collares et al., 2018). The circulation, mixing rates and turbulent processes are poorly explored in the NAP, thus there is a need for further studies on these processes to better comprehend the local water mass transformation in the different depth layers (e.g., Brearley et al., 2017) as it ultimately impacts the physico-chemical properties and biology of

the region. The NAPv1.0 also distinguishes well the difference between the central and eastern basins of the Bransfield Strait (Fig. 10i-l). Colder, denser and more oxygenated deep waters are observed in the former basin, whereas the latter is relatively

warmer, lighter and less oxygenated. This dichotomy occurs because the central basin has restricted connections with the adjacent region due to bathymetric constrains and it also receives more contribution from HSSW than the eastern basin (Gordon et al., 2000; Dotto et al., 2016). Within the Bransfield Strait, more mCDW access the region in autumn coming from Livingston Island and reaching the eastern basin, increasing temperature and salinity in deep waters in all basins (Fig. S4). Moreover, the Peninsula Front migrates toward the South Shetland Islands due to higher inflow of Weddell-sourced waters (Fig. S4e) until

it vanishes in winter due to surface cooling (Fig. S5d). In spring, the hydrographic properties of the Bransfield Strait resemble more those of summer in subsurface levels (Fig. S6). Although the winter DO suggests the presence of deep convection in the eastern basin, the distribution of the isopycnals refutes this idea which reinforces the conclusion of Whitworth et al. (1994) that it is unlikely that deep convection occurs in Bransfield Strait. Due to lack of data from autumn to spring, we will focus on the summer conditions.

We now evaluate the NAPv1.0 in sections cross cutting the WAP, at the WOCE-SR4 West (hereafter, SR4), and along the Scotia-Arc—a route of Antarctic Bottom Water export (Fig. 1a; Franco et al., 2007). At the WAP (Fig. 11a-d), the NAPv1.0 shows high values of $\Theta$ and $S_A$ and lower values of DO, associated with CDW, intruding onto the continental shelf. $\Theta$ as high as 1˚C, sourced from upper CDW, floods the continental shelf in those regions (Moffat and Meredith, 2018), and are heat source for the glacier retreating (Cook et al., 2016). Although the salinity values are high, they are not as high as the

lower branch of CDW offshore (Fig. 11b). The isopycnals offshore tilt upwards near the continental slope, suggesting upwelling of CDW (additionally, several hypotheses have been developed to explain the origin of these warm waters, such as eddies shed by the ACC, current deflection by the cross-shelf-cutting canyons, or advections of the ACC onto the continental shelf; e.g., Prézelin et al., 2000; Martinson et al., 2008; Martinson and McKee, 2012; Couto et al., 2017; McKee and Martinson, 2020b). On the Weddell Sea side, the NAPv1.0 shows the main water masses of the region (Fig. 11e-h), including the Winter

Water at the subsurface, WDW with largest (lowest) values of $\Theta$ and $S_A$ (DO), Weddell Sea Deep Water and Weddell Sea Bottom Water restricted below 4000 m (Farhbach et al., 2004; Kerr et al., 2018c). Another important feature observed in the climatology is the downslope flow of a cold, fresh, dense and recently ventilated waters at the continental slope (van Caspel et al., 2015). At the shelf break, the isotherms tilt suggesting the presence of the Antarctic Slope Front (Farhbach et al., 2004; Heywood et al., 2004). Once the Weddell Sea deep waters enters the Powell Basin, a fraction is exported across the Scotia Arc

(Fig. 1b; Franco et al., 2007). This region is transitional between the Weddell Sea and the northward parts of the Southern Ocean and generally has a warm and weak density structure. The export of dense waters may occur at Philip Passage, where the upper parcel of Weddell Sea Deep Water can overflow the bathymetric constraint, as shown in Fig. 11i-l. On seasonal time scales, the NAPv1.0 represents the natural intraseasonal variability in the upper ocean as well as changes in the isopycnals tilting, suggesting small changes in the circulation across the sections. However, mainly in winter, the data is spatially sparse.

For all seasons except summer, DO distribution is flawless (Figs. S7-S9).

## 4.3 Comparison against other climatological products

Most climatological products are built to represent the global ocean and its large-scale basins, such as the Southern Ocean. The NAP, on the other hand, is a relatively small region but highly dynamical and ecologically important to connect regional environments (Kerr et al., 2018a). In this context, we now compare the NAPv1.0 with some other available gridded products, widely used by the ocean community, to show how effective is to have regional climatologies, especially in areas of intense spatial and temporal variability. For these comparisons, we chose to use the WOA and CARS climatologies and the SOSE state estimate previously described in Sect. 2. Although SOSE is based on a model, and one could argue that comparing it with a gridded product is not a fair comparison, we decided to used it because the data coverage of the Southern Ocean is relatively poor in space and time, and many studies have used SOSE as a benchmark to evaluate other ocean model outputs (e.g., Spence et al., 2017; Russell et al., 2018). In addition, SOSE has been used as initial conditions for simulations within the Bransfield Strait (e.g., Zhou et al., 2020). Despite the limitations of state estimate and ocean reanalysis (e.g., Mazloff et al., 2010; Azaneu et al., 2014; Dotto et al., 2014; Aguiar et al., 2017; Verdy and Mazloff, 2017), combining model and observations, such as in SOSE, leads to great improvements on the comprehension of the Southern Ocean dynamics (e.g., van Sebille et al., 2013; Abernathey et al., 2016; Rodriguez et al., 2016; Tamsitt et al., 2017).

The vertical water mass structure of the NAP and adjacent regions is well represented by the NAPv1.0 (Fig. 12a,b). The density distribution and thermohaline ranges agree well with the CTD casts (closer 6 km from the grid points), as previously shown (Sect. 4.1). The summer-averaged WOA is colder and saltier at the surface, but the intermediate to deep ocean seems to have thermohaline range and density structure similar to observations (Fig. 12c,d). The summer-estimated CARS seems to represent well the NAP region, despite being a product with coarser resolution. It has the range of $\Theta$ and $S_A$ in agreement with the observations throughout the water column (Fig. 12e). A good agreement is also observed for the dense waters, where CARS has similar levels of $\gamma^n$ compared to the measured data (Fig. 12f). Contrary to WOA, the NAPv1.0 and the CARS climatology do not show supercooled waters centred at $\gamma^n$ ~28.27 kg m$^{-3}$ (and $\Theta$~2°C), in agreement with the observations (Fig. 12b,d,f). The 2008-2018 summer-averaged SOSE represents the upper ocean considerably fresher, whereas the water masses with $\gamma^n < 27.6$ kg m$^{-3}$ are colder than the observations (Fig. 12g). The dense water masses ($\gamma^n > 28.27$ kg m$^{-3}$) are not well represented in SOSE for the region analysed, being considerably warmer and saltier than the observations (Fig. 12h). However, due to higher temperatures, lighter waters flood the bottom layers of SOSE. The similarity of the NAPv1.0 and the WOA and CARS climatologies to the observations is because these data sets are totally fed by in situ data, whereas SOSE is based on a model run fed by in situ data.

Now, to present the water mass structure, we show the representation of these products along the several sections at the NAP and adjacent regions. We start showing the representation of the climatologies in the central area of the NAP, i.e., the Bransfield Strait, because it has a transitional regime between cold and warm conditions, and it is highly dynamical. WOA unveils the upper ocean considerably colder, and despite having similar ranges of $\Theta$ as the NAPv1.0 in the mid to deep ocean, the distinction between the Bransfield Strait's central and eastern basins is not well represented (Figs. 13a and Figs. S10a).

The $S_A$ distribution in WOA is slightly saltier than the NAPv1.0, and it does not show a clear spatial variability (Fig. 13b). Consequently, the WOA is generally denser than NAPv1.0, mainly in the eastern basin (Figs. 13c and S11a). The $\Theta$ climatology in CARS is in closer agreement with the NAPv1.0, both in the upper ocean and deeper layers (Fig. 13d). Although the deep $\Theta$ is not as low as the NAPv1.0 (Fig. S10g-i), a clear thermal distinction is observed between the central and eastern basin of the Bransfield Strait. On the other hand, the $S_A$ field is slightly saltier in CARS than the NAPv1.0, and the spatial distribution is less variable, as observed by the flatness of the isohalines (Fig. 13e). In general, CARS is slightly denser than NAPv1.0 for the Bransfield Strait (Fig. S11g). Interesting, however, is the representation of DO in CARS, which resembles the NAPv1.0 in terms of magnitude and spatial distribution (Fig. 13g). In both products, the DO is higher in the central basin (although slightly overestimated in CARS climatology). SOSE is considerably warmer in most of the water column in both basins (Figs. 13h and S10m) when compared to the NAPv1.0 (Fig. 10i). As discussed previously, SOSE shows higher salinity than the NAPv1.0 in both basins below ~250 m depth (Fig. 13i). However, the deep density structure in the Bransfield Strait is lighter than NAPv1.0 because of the higher $\Theta$ values in SOSE (Fig. S11m). Regarding DO, SOSE is oversaturated in the upper layer and highly deoxygenated bellow ~100 m (Fig. 13k), compared to NAPv1.0 (Fig. 10l).

Regarding the other sections in the Bransfield Strait (Figs. 1a), no product captured well the Peninsula and the Bransfield fronts as NAPv1.0, likely due to their coarser resolution (Fig. 14). The mid-depth inflow of mCDW is better represented in CARS, where it is restricted to the South Shetland Islands side. In WOA and SOSE, the mCDW seems to be present in the whole meridional extent of the Bransfield Strait (Fig. 14a,b,f,g,k,l), which creates a band of higher temperatures in levels deeper than 200 m in both products compared to NAPv1.0 (Fig. S10). As previously noticed, the temperature range of CARS is in better agreement with the regional climatology (Fig. S10). In terms of $\gamma^n$, WOA and SOSE are considerably lighter than NAPv1.0 below ~300 m (Fig. S11). For instance, the isopycnals of 28.27 kg m$^{-3}$ and 28.4 kg m$^{-3}$ are found deeper in the sections Livingston-Antarctic Peninsula and King George-Antarctic Peninsula in WOA (Fig. 15a-b). In SOSE, the 28.27 kg m$^{-3}$ and 28.4 kg m$^{-3}$ isopycnals are respectively absence in the former and later sections (Fig. 15k-l). CARS is more in agreement with NAPv1.0 in terms of magnitude and distribution of the isopycnals within the Bransfield Strait despite its larger resolution (Fig. 15f-g). However, in general, the dome-like structure of the isopycnals in the Bransfield Strait, associated with the cyclonic circulation, is not apparent in the other products analysed, which suggest a misrepresentation of the circulation pattern of the strait (Fig. 15).

In the adjacent regions of the NAP, where the oceanographic regime is less dynamical, WOA, CARS and SOSE represent the vertical structure of the water masses relatively better than within the central NAP (Figs. 14 and 15). All of them successfully represent the core of CDW offshore, the upward tilting of the isotherms and isopycnals towards the continental slope and the CDW accessing the WAP continental shelf (Figs. 14 and 15). In the Weddell Sea, all products represent the core of the WDW and the thermal and density distinction of Weddell Sea Deep Water and Weddell Sea Bottom Water. However, only CARS can represent the downslope flow of cold and dense waters at the continental slope, in agreement with NAPv1.0 and observations (Figs. 14 and 15). Along the Scotia-Arc, despite all products roughly agree regarding the distribution of the

warmer water parcel (Fig. 14), only CARS and SOSE represent the export of lighter Weddell Sea Deep Water through Philip Passage, in agreement with NAPv1.0 and observations (Fig. 15).

One must keep in mind that WOA and CARS are global climatologies, and their relatively low-resolution (i.e. 1/4˚ and 1/2˚, respectively) does not represent properly some local and regional environments of the Southern Ocean, like in the NAP (for instance, the distance between the South Shetland Islands and the Antarctic Peninsula is slightly more than 100 km). Despite having lower resolution, CARS seemed to represent better the hydrographic properties in the Bransfield Strait than WOA. Although beyond the scope of this work, this discrepancy could be associated with the interpolation methods that are

distinct for each climatology. On the other hand, SOSE is a state estimate where a model is constrained by observations. Although its output products are good to study and represent large scale processes, SOSE did not seem fit to be used in regional seas, at least in the NAP, where complex dynamics and contrasting regimes set the local water mass structure and variability. In this sense, simulations of regional models could be significantly impacted by the use of those large scale climatologies and reanalysis products.

### 4.4. Caveats of the NAPv1.0 climatology

        In summary, the NAPv1.0 is robust in the representation of many described dynamic features of the region and, to our knowledge, its hydrographic fields closely follow the observations better than any other available product. Hence, the NAPv1.0 can be used to represent the mean state conditions as well as to feed ocean regional models of the highly transitional

and dynamic NAP environments. However, a few caveats can be identified in the NAPv1.0.

        In situ data are spatially and temporally sparse in the Southern Ocean. The Argo and MEOP data sets could have the potential to overcome this uneven sampling, however a few regions are still under sampled, such as the western Weddell Sea (Fig. 2). To avoid creating larger errors (and unreal hydrographic values) in the interpolation procedure, we decided to avoid those regions by masking the final products.

CTD data is still the only data set that can measure deeper (>2000 m) levels in the region. Considering that most of the hydrographic cruises were conducted in summer (Fig. 3), deeper regions are not well represented in other seasons. In those cases, most of the climatology is developed to the upper ocean (<2000 m) given the measurement restrictions of the Argo and MEOP datasets. To avoid creating larger errors (and unreal hydrographic values) in the interpolation procedure, we decided not to interpolate where data is not available. Because of this choice, autumn, winter and spring showed a lower

representativeness than summer. This might be solved in the near future with the development of deep-Argo floats (Roemmich et al., 2019), and could potentially sample many deep parts of the adjacent regions of the NAP (e.g., Weddell Sea, Bellingshausen Sea and Drake Passage).

        Another limitation of our climatology is regarding the DO field. As most of the CTD measurements were conducted in summer, the greatest representation of this filed is restricted to that season. In addition, a considerable amount of DO

measurements was conducted by the GOAL, restricted to the Bransfield and Gerlache Straits during summer (Fig. 3c). The

development of biogeochemical Argo floats could increase the amount of available DO data in the future for different the seasons (Roemmich et al., 2019), which would improve the representation of the different climatologies.

The grid scale (~10 km) is similar or even higher than the first baroclinic Rossby radius of the region (Chelton et al., 1998), which limits the representation of some meso scale and submesoscale features. In addition, the smoothing radius chosen in the objective interpolation and the mean state associated with the interpolation filter out most of the small dynamic features, such as eddies associated with the Peninsula Front (e.g., Sangrà et al., 2011) or at the WAP continental slope (Martinson and McKee, 2012).

Finally, the NAPv1.0 product brings a novel tool and opens plenty of possible future applications to better understand the regional circulation and hydrography along the NAP, a recognized marine climate hotspot (Kerr et al., 2018a). This climatology will be useful not only to modellers, but also to ecologist and anyone interested in the NAP region.

## 5 Data availability

The seasonal NAPv1.0 is available at https://www.goal.furg.br/producao-cientifica/supplements/203-goal-gridded-nap and at https://doi.org/10.5281/zenodo.4420006 (Dotto et al., 2021), as a netCDF file. The file contains the 3D gridded variables in situ temperature (˚C), practical salinity, and dissolved oxygen (mL L$^{-1}$), and their respective relative error matrices. Derived variables, such as absolute salinity, conservative temperature and neutral density can be calculated by the user. Depth levels (in metres) and the 2D gridded coordinates latitude and longitude are also included in the file. Missing values corresponding to relative errors > 0.5 and land mask are marked as *not a number* (NaN).

## 6 Conclusions

Here, we presented a novel gridded hydrographic product for the NAP and adjacent regions generated by optimal interpolation using hydrographic data from CTD, Argo and MEOP between 1990-2019. The gridded product has a spatial resolution of ~10 km and 90 standard depths with spacing ranging from 5 m in the upper ocean to 500 m in depths >4000 m. The NAPv1.0 represents quite well many oceanic features of the region, such as (i) the inflows of Weddell Sea-sourced waters and the warm regime from the Bellingshausen Sea, (ii) the Peninsula and the Bransfield Fronts, (iii) the cyclonic circulation within the Bransfield Strait, (iv) the Bransfield Strait central and eastern basins dichotomy hydrographic regime, (v) the inflow of CDW onto the WAP continental shelf, (vi) the narrow flow of cold and oxygen-rich water along the continental slope of the Weddell Sea inflowing toward the Bransfield Strait, and (vii) the downslope flow of recently ventilated dense bottom waters at the Weddell Sea continental slope. Many of the features described are not well observed in some of the other products evaluated and depicting the mechanisms behind those differences still poses a challenge to the community. Due to the larger mapping errors, one must have caution when considering areas near the edge of the NAP and as well as areas of limited data coverage. As caveats, the NAPv1.0 is fed exclusively by in situ data and the reduced sampling in deeper levels and seasons other than summer limits the representation of many oceanographic features for autumn, winter and spring. Nevertheless, the

NAPv1.0 is a valuable tool for setting up regional ocean models and ocean reanalysis assessment, or to any user who wants to use it to characterize the mean-state hydrography of the NAP in the end of 20th-century and early 21st-century.

**Author contribution.** All authors designed and conceptualized the study. TSD led the study, data processing, and writing of the manuscript. RK, MMM, and CAEG lead the GOAL projects and contributed substantially to writing the manuscript.

**Competing interests.** The authors declare that they have no conflict of interest.

**Acknowledgement.** This study is part of the activities of the Brazilian High Latitude Oceanography Group (GOAL) within the Brazilian Antarctic Program (PROANTAR). GOAL has been funded by and/or has received logistical support from the Brazilian Ministry of the Environment (MMA), the Brazilian Ministry of Science, Technology, and Innovation (MCTI), the Council for Research and Scientific Development of Brazil (CNPq), the Brazilian Navy, the Inter-Ministerial Secretariat for Sea Resources (SECIRM), the National Institute of Science and Technology of the Cryosphere (INCT CRIOSFERA; CNPq grants No. 573720/2008-8 and 465680/2014-3) and the Research Support Foundation of the State of Rio Grande do Sul (FAPERGS grant No. 17/2551-000518-0). This study was conducted within the activities of the REDE-1, SOS-CLIMATE, POLARCANION, PRO-OASIS, NAUTILUS, INTERBIOTA, PROVOCCAR and ECOPELAGOS projects (CNPq grants No. 550370/2002-1, 520189/2006-0, 556848/2009-8, 565040/2010-3, 405869/2013-4, 407889/2013-2, 442628/2018-8 and 442637/2018-7, respectively). Financial support was also received from Coordination for the Improvement of Higher Education Personnel (CAPES) through the project CAPES "Ciências do Mar" (grant No. 23038.001421/2014-30). CAPES also provided free access to many relevant journals though the portal "Periódicos CAPES", and the activities of the Graduate Program in Oceanology. T.S. Dotto acknowledge financial support from CNPq PDJ scholarship grant No. 151248/2019-2. R. Kerr, M. M. Mata and C. A. E. Garcia are granted with researcher fellowships from CNPq grants No. 304937/2018-5, 306896/2015-0, 309932/2019-0, respectively. The authors thank to the officers and crew of the polar vessels *Ary Rongel* and *Almirante Maximiano* of the Brazilian Navy, and the several scientists and technicians participating in the cruises for their valuable help during data sampling and data processing. We also thank the efforts all scientists, technicians and crew involved in all data acquisition/processing of ship surveys, and Argo and MEOP programs, and for making them freely available in international repositories. Argo data were collected and made freely available by the International Argo Program and the national programs that contribute to it (http://www.argo.ucsd.edu, http://argo.jcommops.org). The Argo Program is part of the Global Ocean Observing System. The marine mammal data were collected and made freely available by the International MEOP Consortium and the national programs that contribute to it. (http://www.meop.net). Finally, we thank Hartmut H. Hellmer and one anonymous reviewer for their contributions to improve the manuscript.

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

**Table 1.** Standard depth levels (m) used to linearly interpolate the profiles and the gridded data set.

| Level number | Depth interval | Depth range |
|:---:|:---:|---:|
| 1 to 10 | 5 m | 5-50 m |
| 11 to 30 | 10 m | 60-250 m |
| 31 to 52 | 25 m | 275-800 m |
| 53 to 66 | 50 m | 850-1500 m |
| 67 to 81 | 100 m | 1600-3000 m |
| 82 to 86 | 200 m | 3200-4000 m |
| 87 to 90 | 500 m | 4500-6000 m |

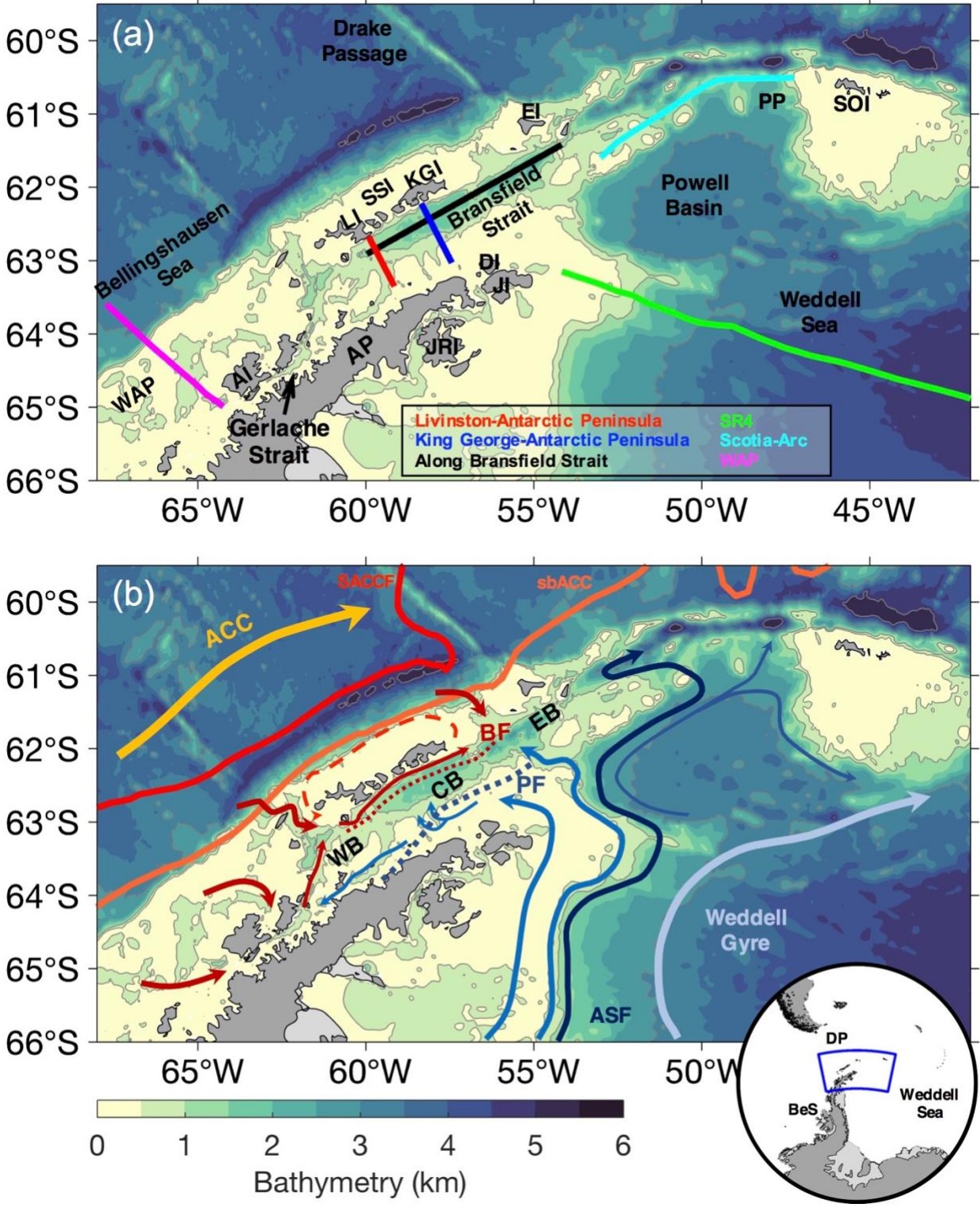

**Figure 1.** (a) Bathymetry of the study region. South Shetland Islands (SSI), Antarctic Peninsula (AP), Elephant Island (EI), Joinville Island (JI), D'Urville Island (DI), James Ross Island (JRI), King George Island (KGI), Livingston Island (LI), Anvers Island (AI), Philip Passage (PP) and South Orkney Island (SOI) are shown. The coloured lines depict the sections used to analyse the NAPv1.0 climatology (see legend). Bathymetry from ETOPO1. Isobaths of 500, 1000, 3000 and 5000 m are shown in grey line. (b) Schematic of the mean circulation at the Northern Antarctic Peninsula (NAP). Blue (red) arrows shows the cold (warm) water masses that inflow in the region. Cyan arrow shows

the Weddell Gyre circulation and the dark blue arrow the path of the Antarctic Slope Front (ASF). Dotted blue and red lines depict, respectively, the Peninsula Front (PF) and the Bransfield Front (BF). Dashed line shows the recirculation around the SSI. The Antarctic Circumpolar Current (ACC), the Southern ACC Front (SACCF) and the Southern Boundary of the ACC (sbACC), are shown in orange tones. Within the Bransfield Strait, the area is divided in eastern basin (EB), central basin (CB) and western basin (WB). Inset shows the study region, among the Weddell Sea, Bellingshausen Sea (BeS) and the Drake Passage (DP). Coastline from SCAR Antarctic Digital

Database.

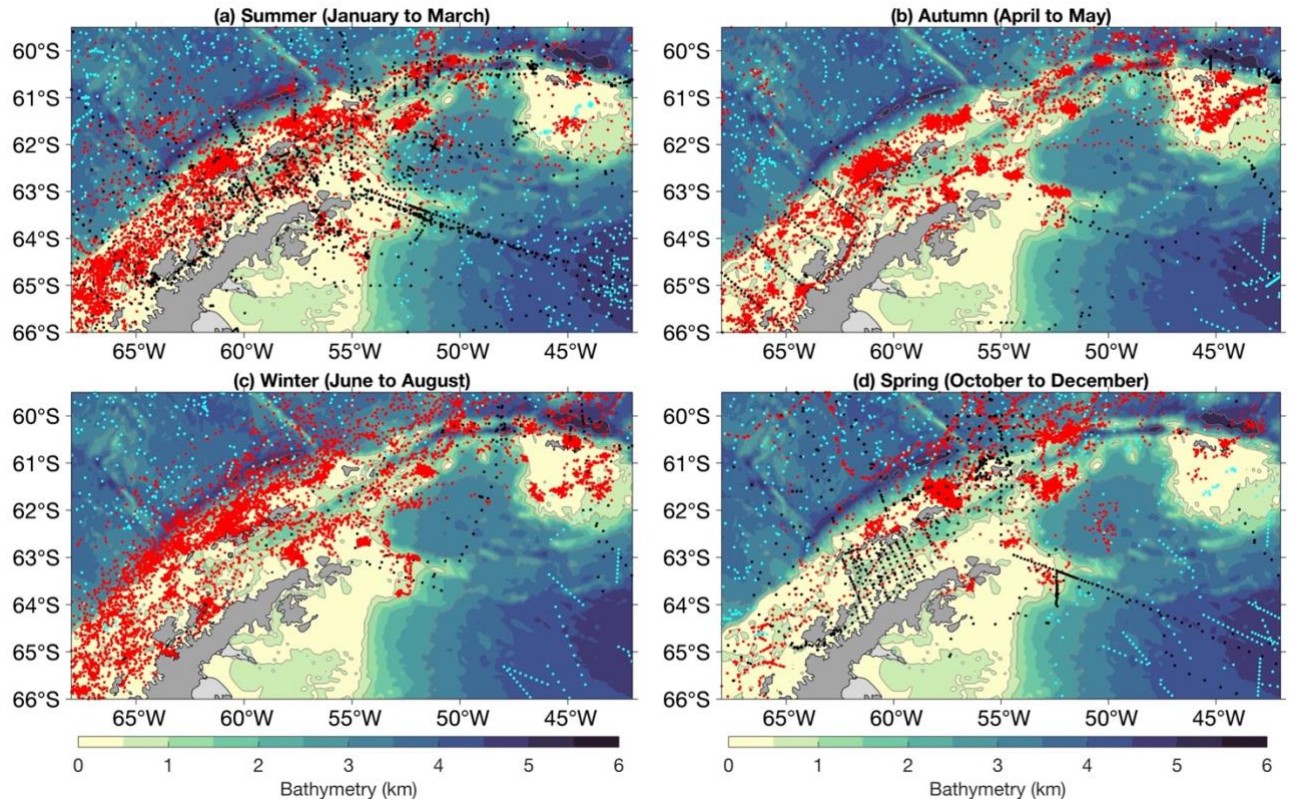

**Figure 2.** Distribution of the hydrographic data sets used to build the NAPv1.0 climatology in (a) summer, (b) autumn, (c) winter and (d) spring seasons. CTD (black), MEOP (red) and Argo (cyan) casts are shown. The data set was restricted to 1990 to 2019. ETOPO1 isobaths of 500 m, 1000 m, 3000 m and 5000 m are depicted by grey lines. Coastline from SCAR Antarctic Digital Database.

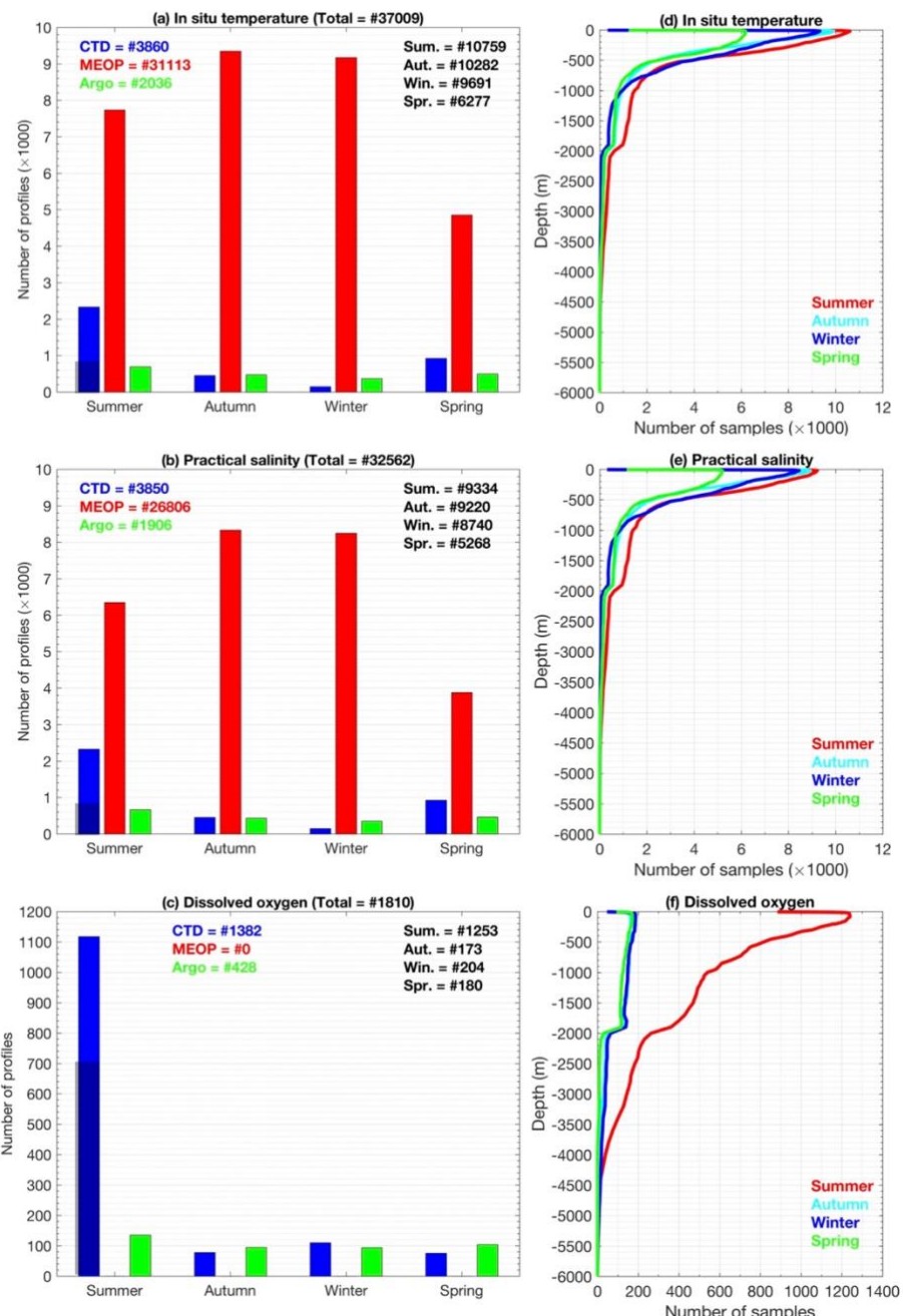


**Figure 3.** (a) Number of in situ temperature profiles per season and platform (CTD in blue, MEOP in red and Argo in green) for the study region. (b) Same as panel (a), but for practical salinity. (c) Same as panel (a), but for dissolved oxygen. MEOP platforms do not carry oxygen sensors. The black shading in summer bars show the contribution of GOAL profiles. The total number of profiles in shown in each panel title, the total number of profiles per season is shown in black in the right upper corner of each panel and the total number of profiles per

platform in shown by colours in the left upper corner of each panel. Note the different y-axis scales for the dissolved oxygen. (d) Total number of samples per depth for in situ temperature considering all platforms in summer (red), autumn (cyan), winter (blue) and spring (green). (e) Same as panel (d), but for practical salinity. (f) Same as panel (d), but for dissolved oxygen.

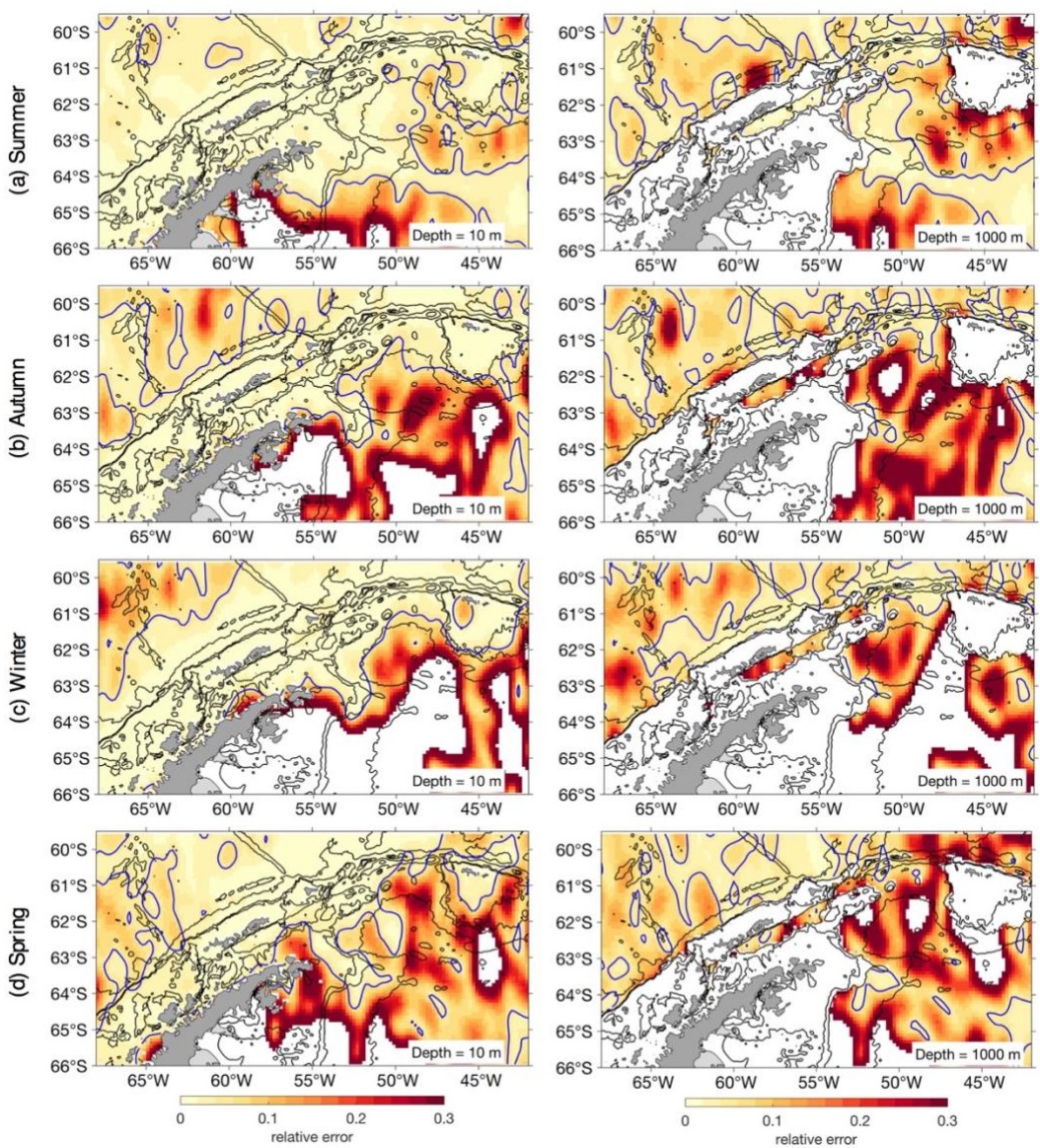


**Figure 4.** Relative error for in situ temperature calculated in the objective interpolation at 10 m (left panels) and 1000 m (right panels) depth for (a) summer, (b) autumn, (c) winter and (d) spring. Relative error of 0.05 is marked by the blue line. Areas where the relative error is > 0.5 are blank. ETOPO1 isobaths of 500 m, 1000 m, 3000 m and 5000 m are depicted by black lines. Coastline from SCAR Antarctic Digital Database.

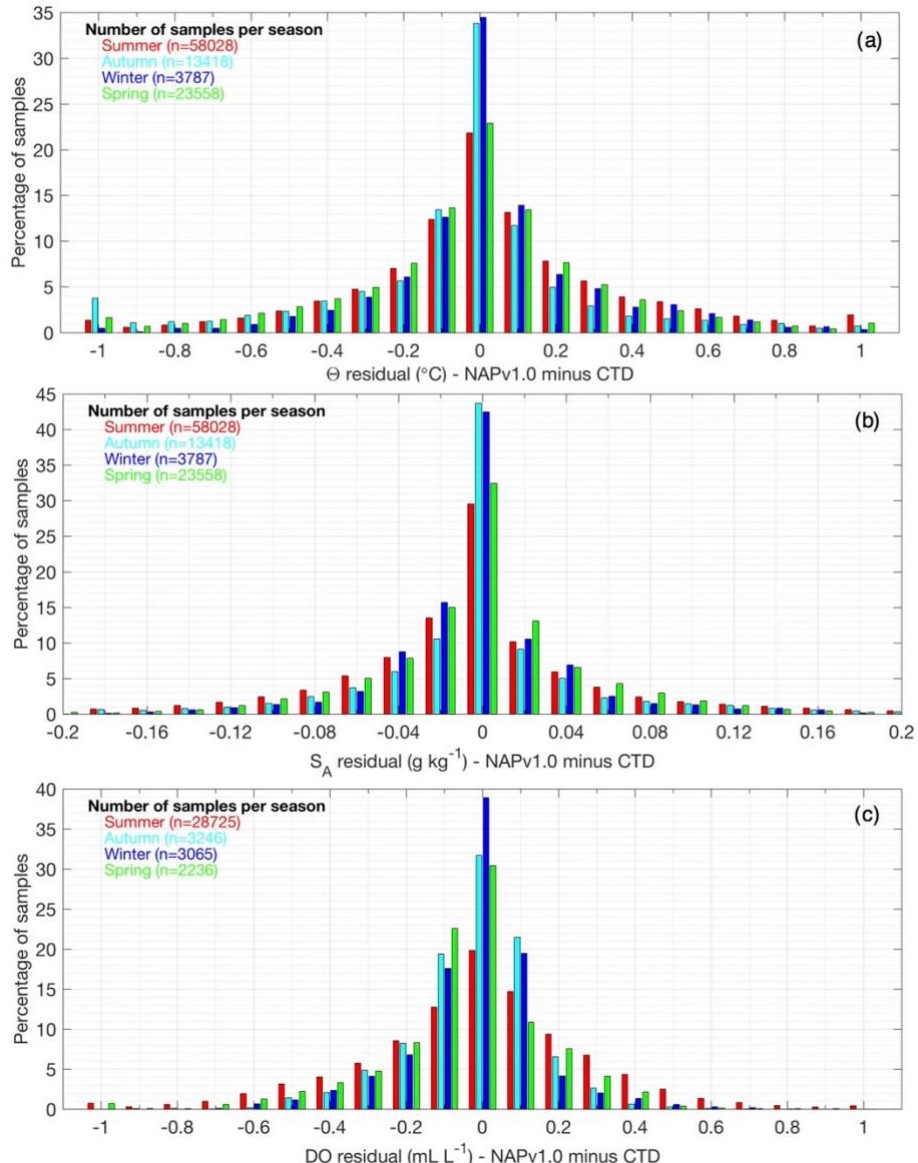

**Figure 5.** (a) Histogram showing the percentage of the residual difference of conservative temperature (Θ; ˚C) samples between the NAPv1.0 and the CTDs closer ~6 km of each grid point at 0.1˚C bins interval per season. The seasonal difference is show by colours: summer (red), autumn (cyan), winter (blue) and spring (green). Number of samples per season is shown in the upper left corner of the panel according to the respective colours. (b) Same as panel (a), but for absolute salinity ($S_A$; g kg$^{-1}$) at 0.02 g kg$^{-1}$ bins. (c) Same as panel (a), but for dissolved oxygen (DO; mL L$^{-1}$) at 0.1 mL L$^{-1}$ bins.

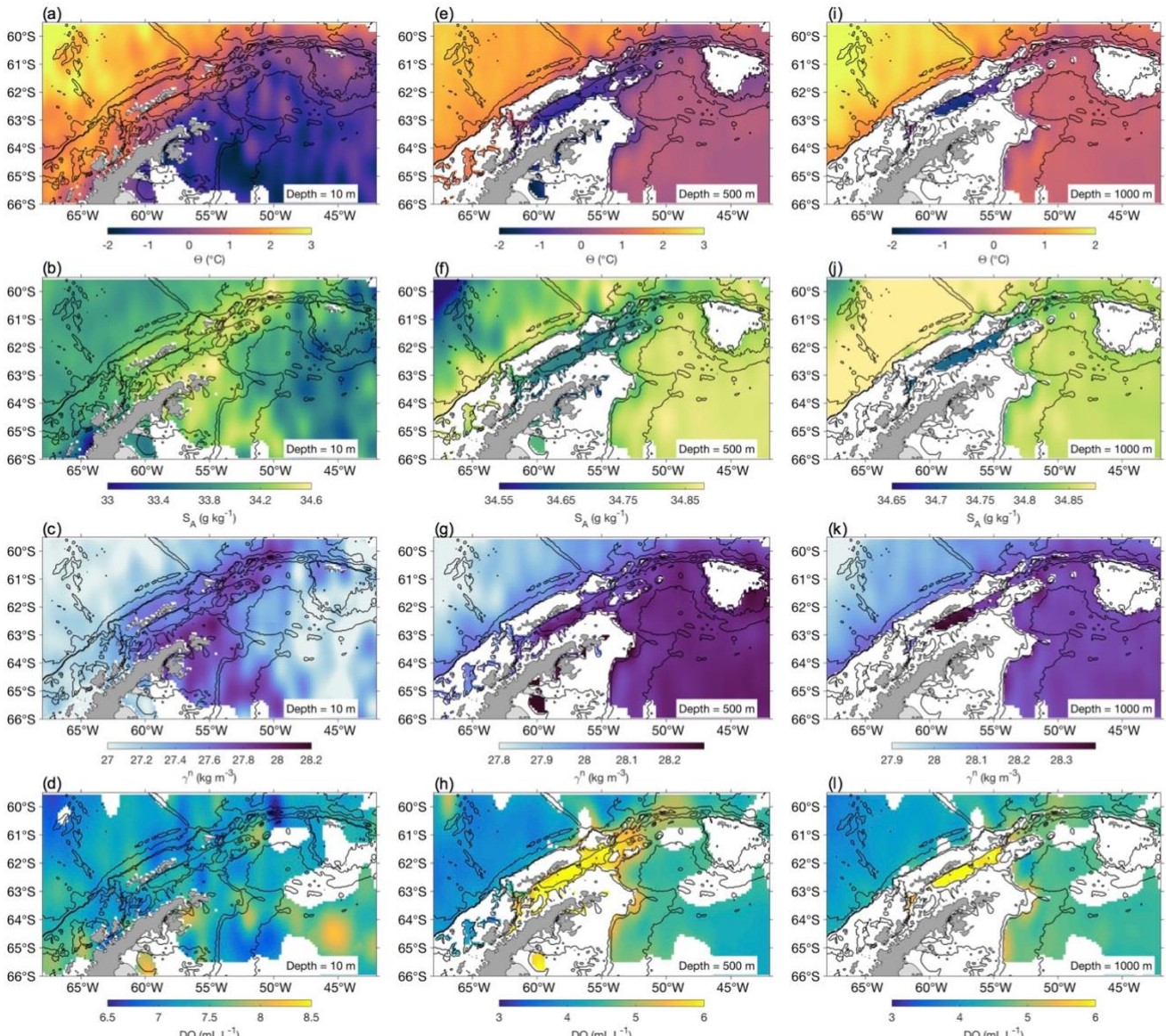

**Figure 6.** Surface maps of NAPv1.0 at 10 m (left panels), 500 m (middle panels) and 1000 m (right panels) depth for summer. (a, e, i) Conservative temperature ($\Theta$; ˚C). (b, f, j) Absolute salinity ($S_A$, g kg$^{-1}$). (c, g, k) Neutral density ($\gamma^n$; kg m$^{-3}$). (d, h l) Dissolved oxygen (DO; mL L$^{-1}$). ETOPO1 isobaths of 500 m, 1000 m, 3000 m and 5000 m are depicted by the black lines. Coastline from SCAR Antarctic Digital Database.


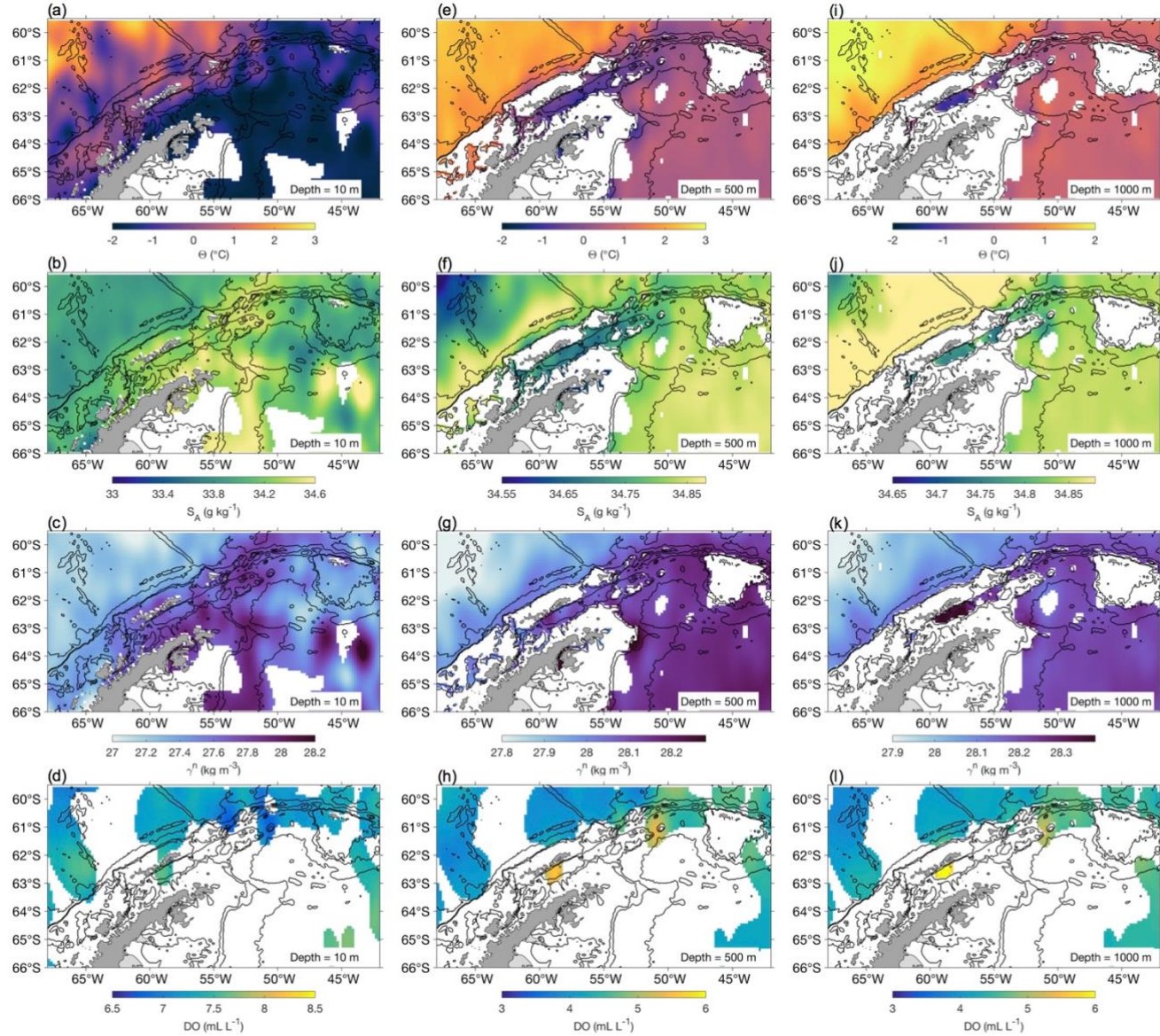

**Figure 7.** Same as Fig. 6, but for autumn.

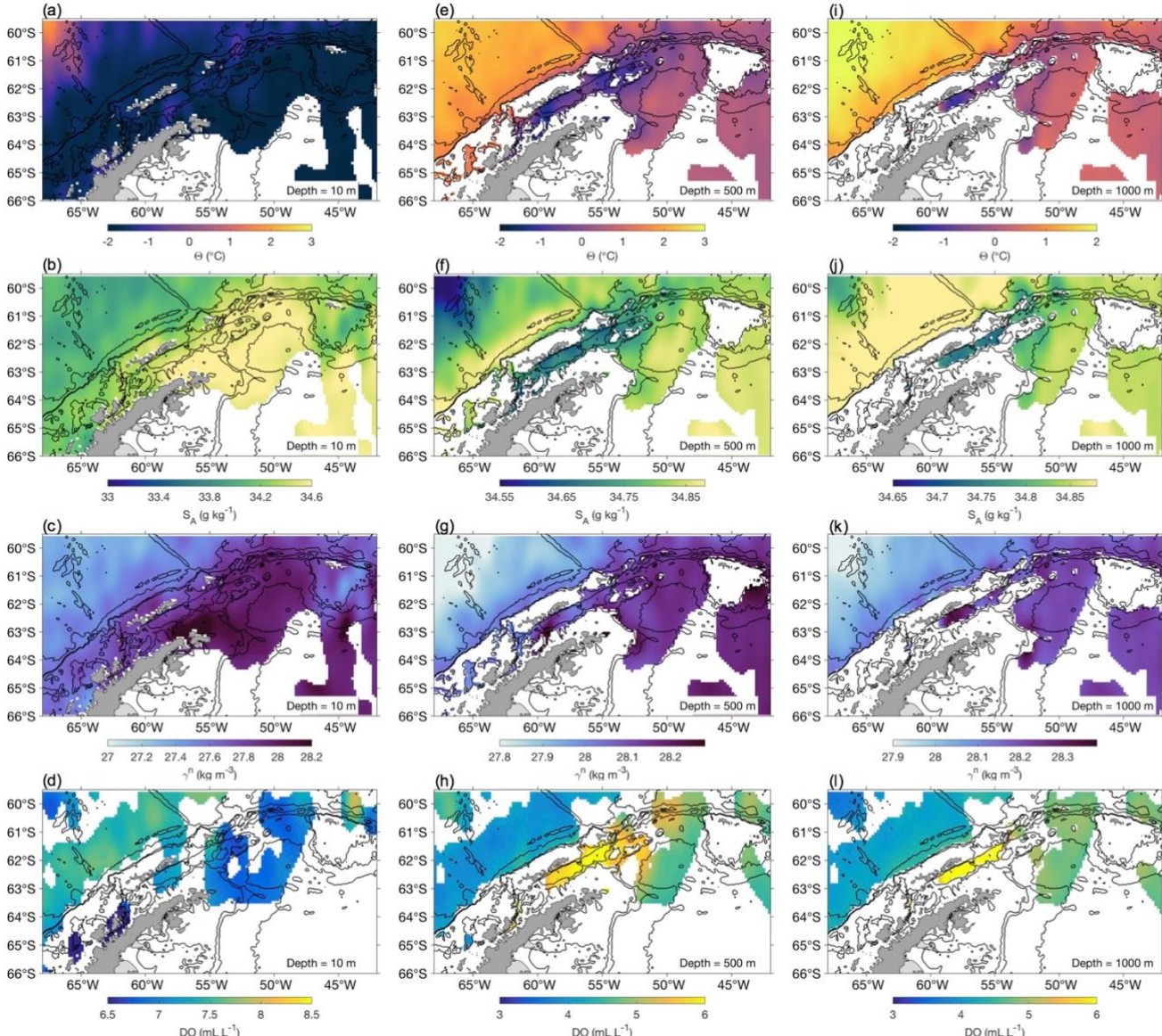

**Figure 8.** Same as Fig. 6, but for winter.

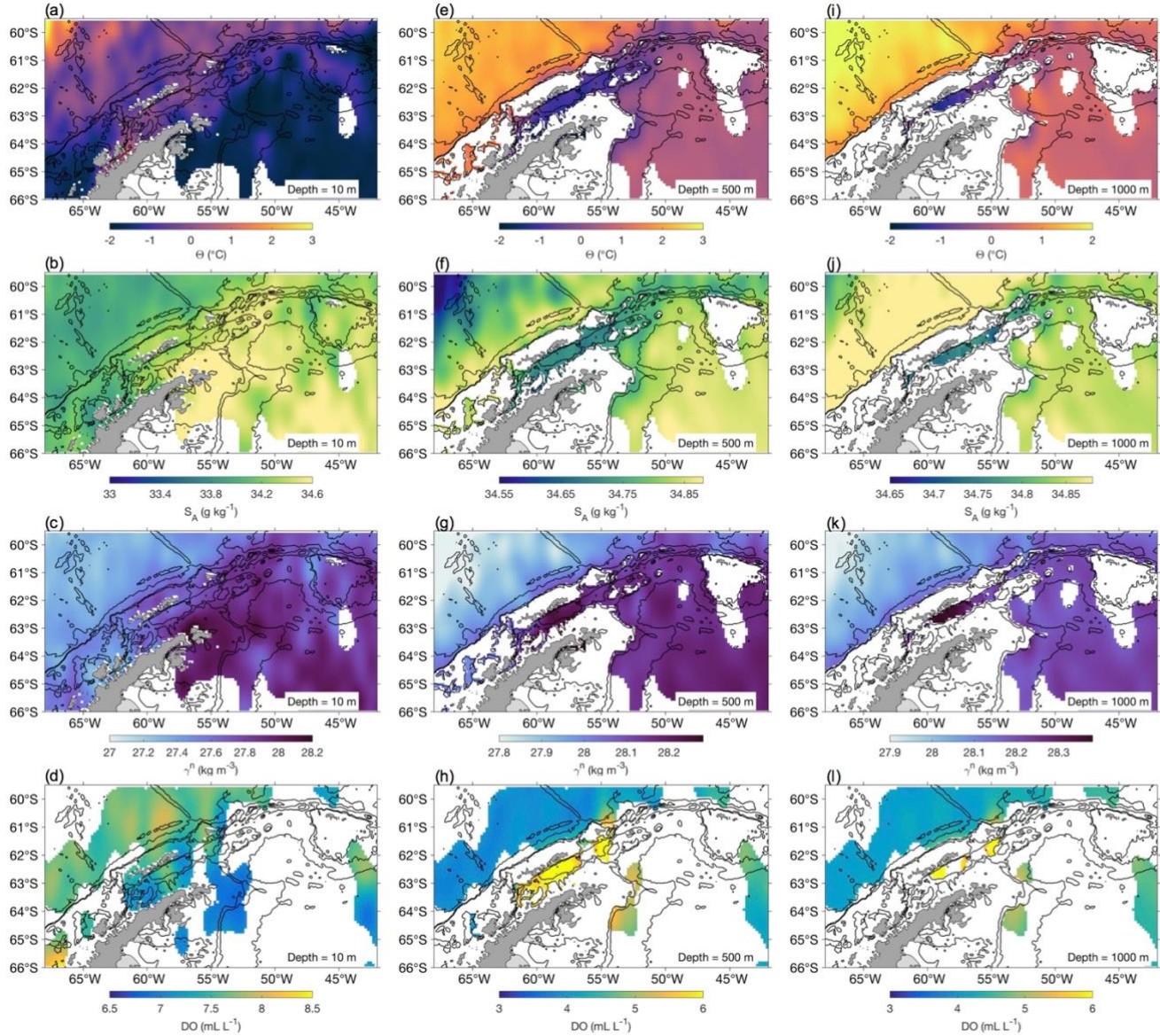

**Figure 9.** Same as Fig. 6, but for spring.

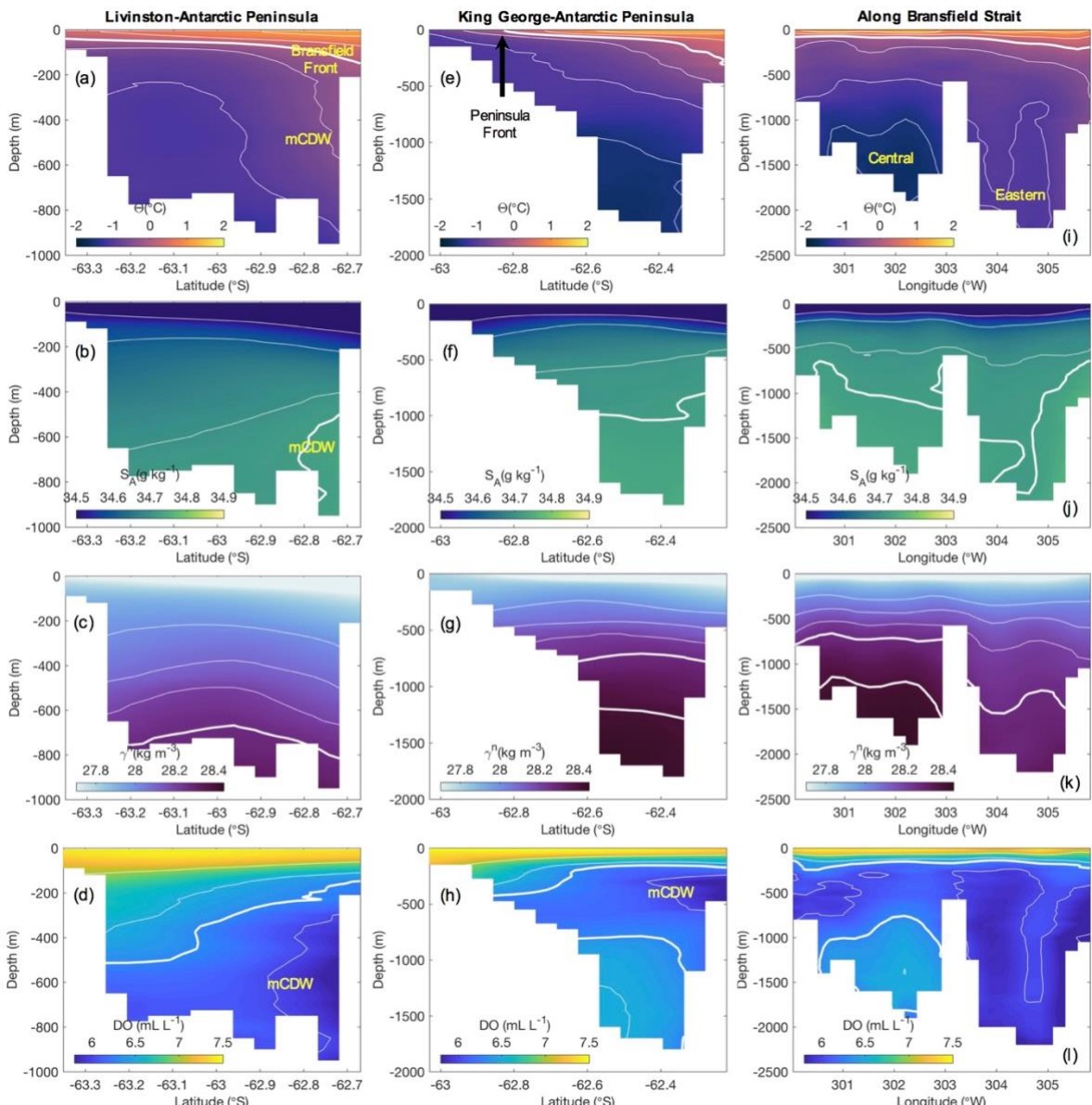

**Figure 10.** Summer vertical sections crossing the Bransfield Strait between the Livingston Island and the Antarctic Peninsula (a-d; red line in Fig. 1a), King George Island and the Antarctic Peninsula (e-h; blue line in Fig. 1a), and along the Bransfield Strait (i-l; black line in Fig. 1a). Conservative temperature ($\Theta$; ˚C) is shown in panels (a), (e) and (i). Isotherm of 0˚C is shown by the thick white line. Thin lines show the isotherms of $-1.5$˚C to $+1.5$˚C every 0.5˚C. Absolute salinity ($S_A$; g kg$^{-1}$) is shown in panels (b), (f) and (j). Isoline of 34.72 g kg$^{-1}$ is shown by the thick white line. Thin lines show the isolines of 34.5 and 34.7 g kg$^{-1}$ every 0.1 g kg$^{-1}$. Neutral density ($\gamma^n$; kg m$^{-3}$) is shown in panels (c), (g) and (k). The isolines of 28.27 and 28.40 kg m$^{-3}$ are shown by the thick white lines. Thin lines show the isolines of 28.00 and 28.20 kg m$^{-3}$ every 0.1 kg m$^{-3}$. Dissolved oxygen (DO; mL L$^{-1}$) is shown in panels (d), (h) and (l). The isolines of 6.3 mL L$^{-3}$ is shown by the thick white lines. Thin lines show the isolines of 5.5 and 7.0 mL L$^{-1}$ every 0.5 mL L$^{-1}$. The Bransfield Front, Peninsula Front and mCDW inflows are identified in panels (a), (b), (d), (e) and (h). The Bransfield Strait central and eastern basins are identified in panel (i). Note the difference in the depth ranges.

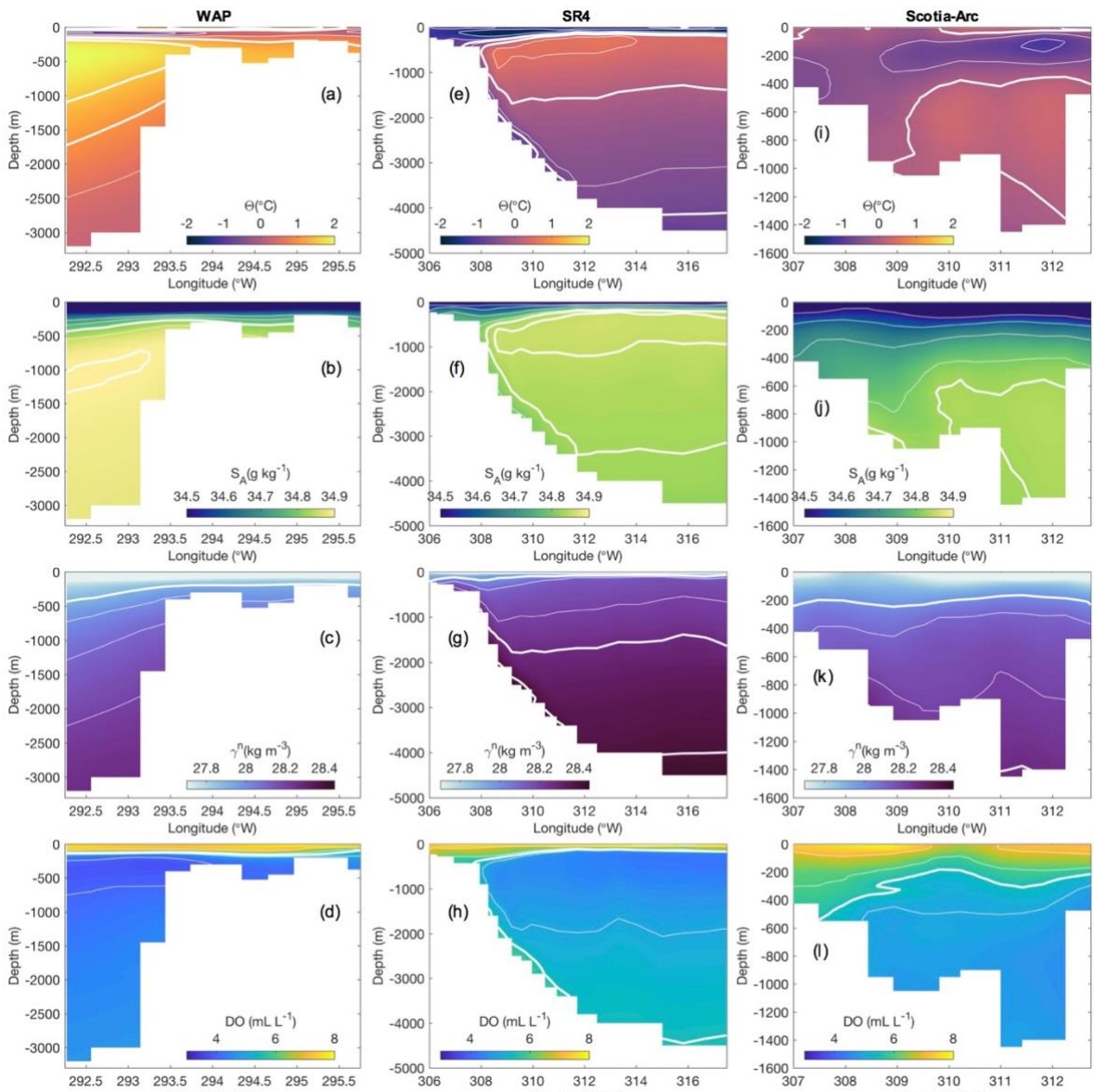

**Figure 11.** Summer vertical sections along the Western Antarctic Peninsula (WAP; a-d; magenta line in Fig. 1a), WOCE SR4 across the Weddell Gyre (e-h; green line in Fig. 1a), and Scotia-Arc line over the South Scotia Ridge (i-l; cyan line in Fig. 1a). (a) Conservative temperature ($\Theta$; ˚C) with isotherms of 1˚C and 1.5˚C (–1.5˚C to 1.5˚C every 0.5˚C) shown by the thick (thin) white line. (b) Absolute salinity ($S_A$; g kg$^{-1}$) with isolines of 34.8 g kg$^{-1}$ and 34.9 g kg$^{-1}$ (34.7 g kg$^{-1}$ to 34.9 g kg$^{-1}$ every 0.05 g kg$^{-1}$) shown by the thick (thin) white line. (c) Neutral density ($\gamma^n$; kg m$^{-3}$) with isoline of 27.9 kg m$^{-3}$ (28.0 kg m$^{-3}$ to 28.2 kg m$^{-3}$ every 0.1 kg m$^{-3}$) shown by the thick (thin) white line. (d) Dissolved oxygen (DO; mL L$^{-1}$) with isoline of 5.5 mL L$^{-1}$ (4 mL L$^{-1}$ to 8 mL L$^{-1}$ every 1 mL L$^{-1}$) shown by the thick (thin) white line. (e) $\Theta$ with isotherms of –0.7˚C and 0˚C (–1.5˚C to 1.5˚C every 0.5˚C) shown by the thick (thin) white line. (f) $S_A$ with isolines of 34.83 g kg$^{-1}$ and 34.85 g kg$^{-1}$ (34.5 g kg$^{-1}$ to 34.8 g kg$^{-1}$ every 0.1 g kg$^{-1}$) shown by the thick (thin) white line. (g) $\gamma^n$ with isolines of 28.0 kg m$^{-3}$, 28.27 kg m$^{-3}$ and 28.4 kg m$^{-3}$ (28.0 kg m$^{-3}$ to 28.2 kg m$^{-3}$ every 0.1 kg m$^{-3}$) shown by the thick (thin) white line. (h) Same as panel (d). (i) $\Theta$ with isotherms of 0˚C (–1.5˚C to 1.5˚C every 0.5˚C) shown by the thick (thin) white line. (j) $S_A$ with isoline of 34.83 g kg$^{-1}$ (34.5 g kg$^{-1}$ to 34.8 g kg$^{-1}$ every 0.1 g kg$^{-1}$) shown by the thick (thin) white line. (k) $\gamma^n$ with isolines of 28.0 kg m$^{-3}$ and 28.27 kg m$^{-3}$ (28.0 kg m$^{-3}$ to 28.2 kg m$^{-3}$ every 0.1 kg m$^{-3}$) shown by the thick (thin) white line. (l) Same as panel (d).

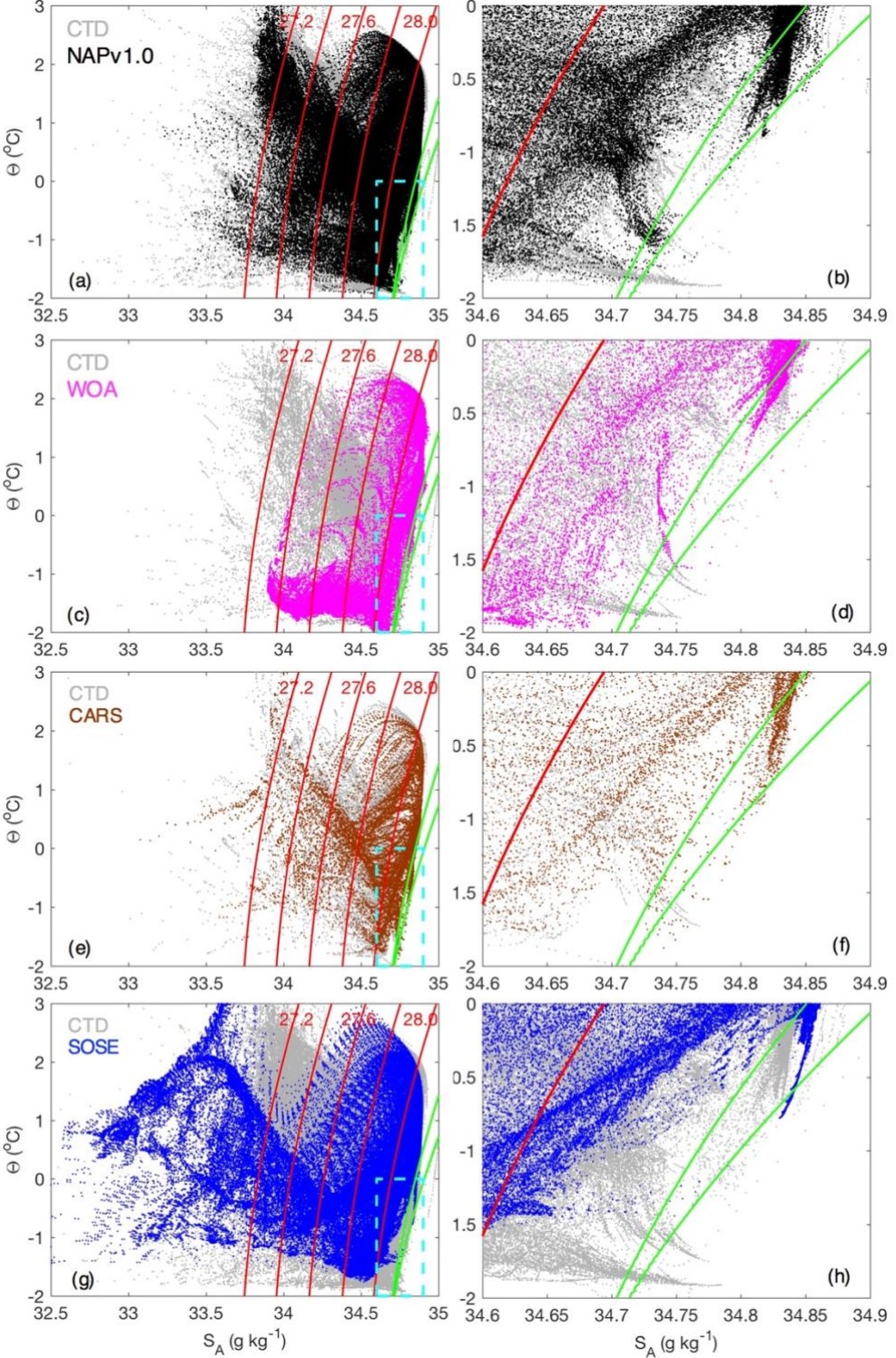

**Figure 12.** Summer conservative temperature-absolute salinity ($\Theta$-$S_A$) diagrams for the different climatologies analysed and profiles collected by CTD (represented by grey points) closer to ~6 km to the grid points. (a and b) NAPv1.0, (c and d) WOA, (e and f) CARS, and (g and h) SOSE. Neutral density isopycnals of 27.2 kg m$^{-3}$ to 28.0 kg m$^{-3}$ are shown every 0.2 kg m$^{-3}$. Green lines depict the 28.27 kg m$^{-3}$ and 28.40 kg m$^{-3}$ isopycnals. Cyan rectangles in (a), (c), (e) and (g) shows the dense water masses restriction presented in (b), (d), (f), and (h), respectively. The difference in the amount of CTD points is because the grid points of each climatology is taken as reference for the stations positions.

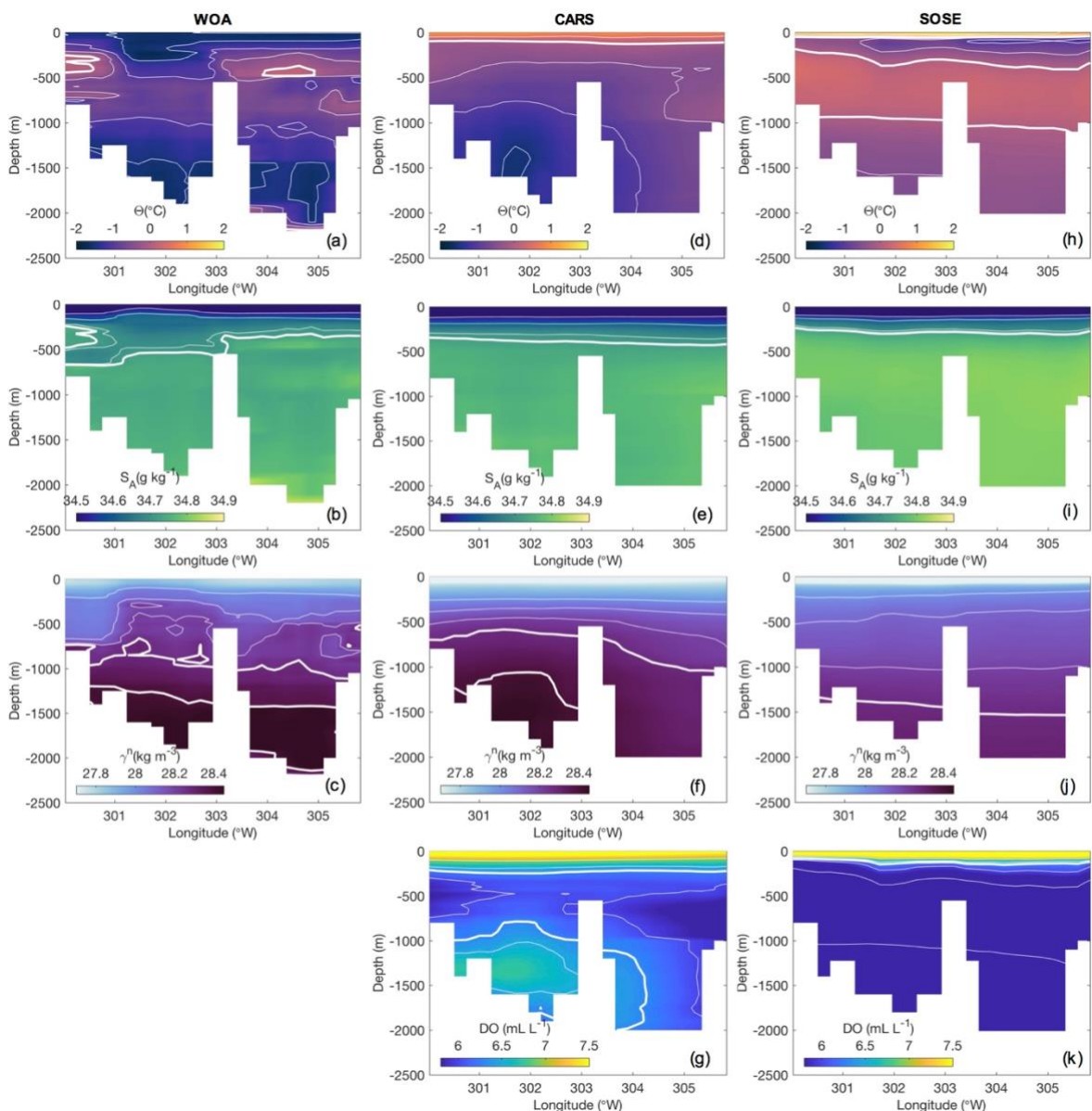

**Figure 13.** Summer vertical sections along the Bransfield Strait from WOA, CARS, SOSE. (a) WOA conservative temperature ($\Theta$; ˚C). Isotherm of 0˚C is shown by the thick white line. Thin lines show the isotherms of $-1.5$˚C to 1.5˚C every 0.5˚C. (b) WOA absolute salinity ($S_A$; g kg$^{-1}$). Isoline of 34.72 g kg$^{-1}$ is shown by the thick white line. Thin lines show the isolines of 34.5 g kg$^{-1}$ and 34.7 g kg$^{-1}$ every 0.1 g kg$^{-1}$. (c) WOA neutral density ($\gamma^n$; kg m$^{-3}$). Isolines of 28.27 kg m$^{-3}$ and 28.4 kg m$^{-3}$ is shown by the thick white line. Thin lines show the

isolines of 28 kg m$^{-3}$ and 28.2 kg m$^{-3}$ every 0.1 kg m$^{-3}$. (d) Same as panel (a), but for CARS. (e) Same as panel (b), but for CARS. (f) Same as panel (c), but for CARS. (g) CARS dissolved oxygen (DO; mL L$^{-1}$). The isolines of 6.3 mL L$^{-1}$ is shown by the thick white lines. Thin lines show the isolines of 5.5 mL L$^{-1}$ and 7.0 mL L$^{-1}$ every 0.5 mL L$^{-1}$. (h) Same as panel (a), but for SOSE. (i) Same as panel (b), but for SOSE. (j) Same as panel (c), but for SOSE. (k) Same as panel (g), but for SOSE.

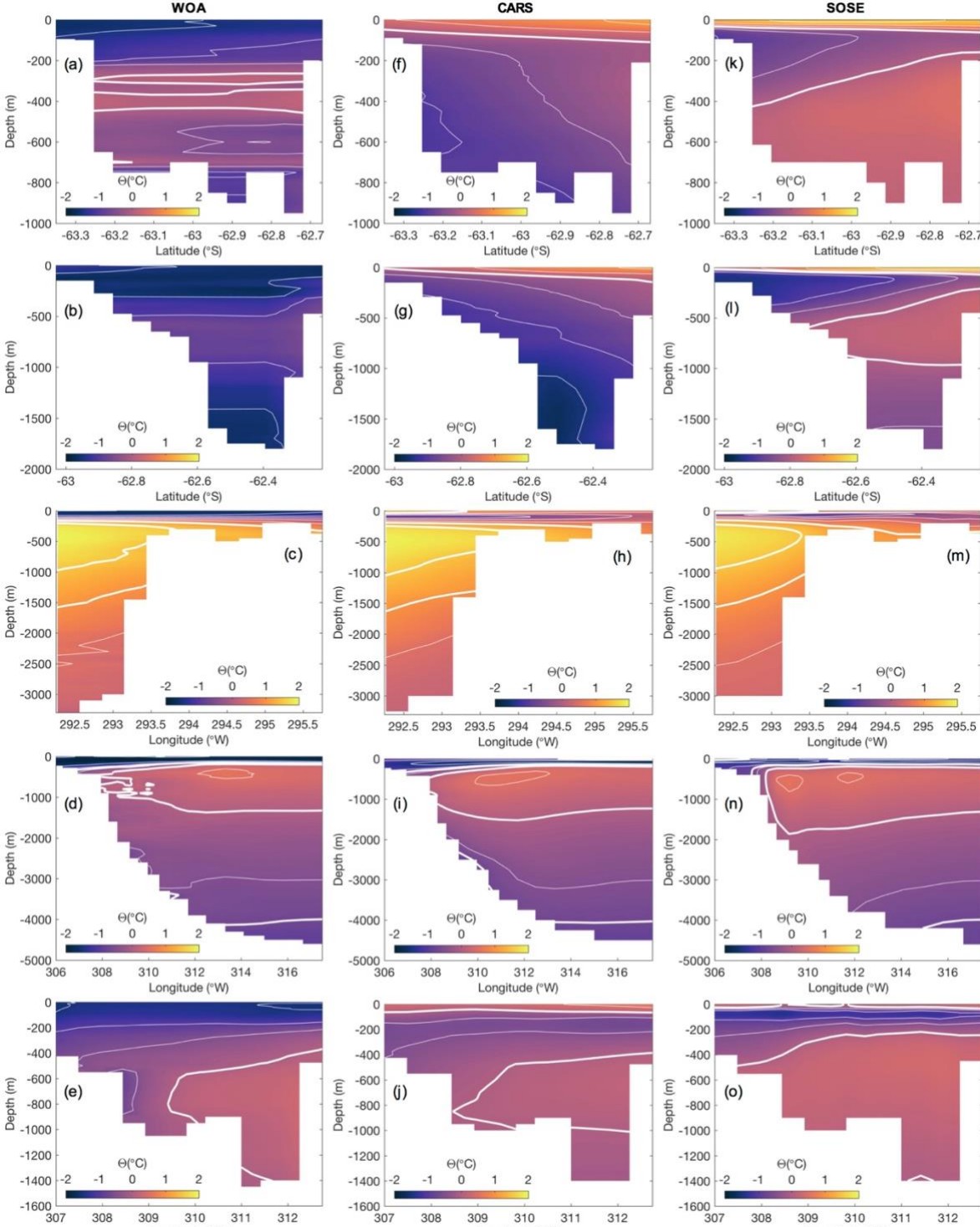

**Figure 14.** Summer vertical sections of conservative temperature (Θ; ˚C) for WOA (left panels), CARS (middle panels) and SOSE (right panels). (a, f, k) Sections between the Livingston Island and the Antarctic Peninsula. Isotherm of 0˚C (−1.5˚C to 1.5˚C every 0.5˚C) is shown by the thick (thin) white line. (b, g, l) Sections between the King George Island and the Antarctic Peninsula. Isotherm of 0˚C (−1.5˚C to 1.5˚C every 0.5˚C) is shown by the thick (thin) white line. (c, h, m) Section at the WAP. Isotherms of 1˚C and 1.5 ˚C (−1.5˚C to 1.5˚C every 0.5˚C) are shown by the thick (thin) white line. (d, i, n) Sections WOCE SR4. Isotherms of −0.7˚C and 0 ˚C (−1.5˚C to 1.5˚C every 0.5˚C) are shown by the thick (thin) white line. (e, j, o) Sections Scotia-Arc. Isotherm of 0 ˚C (−1.5˚C to 1.5˚C every 0.5˚C) is shown by the thick (thin) white line.

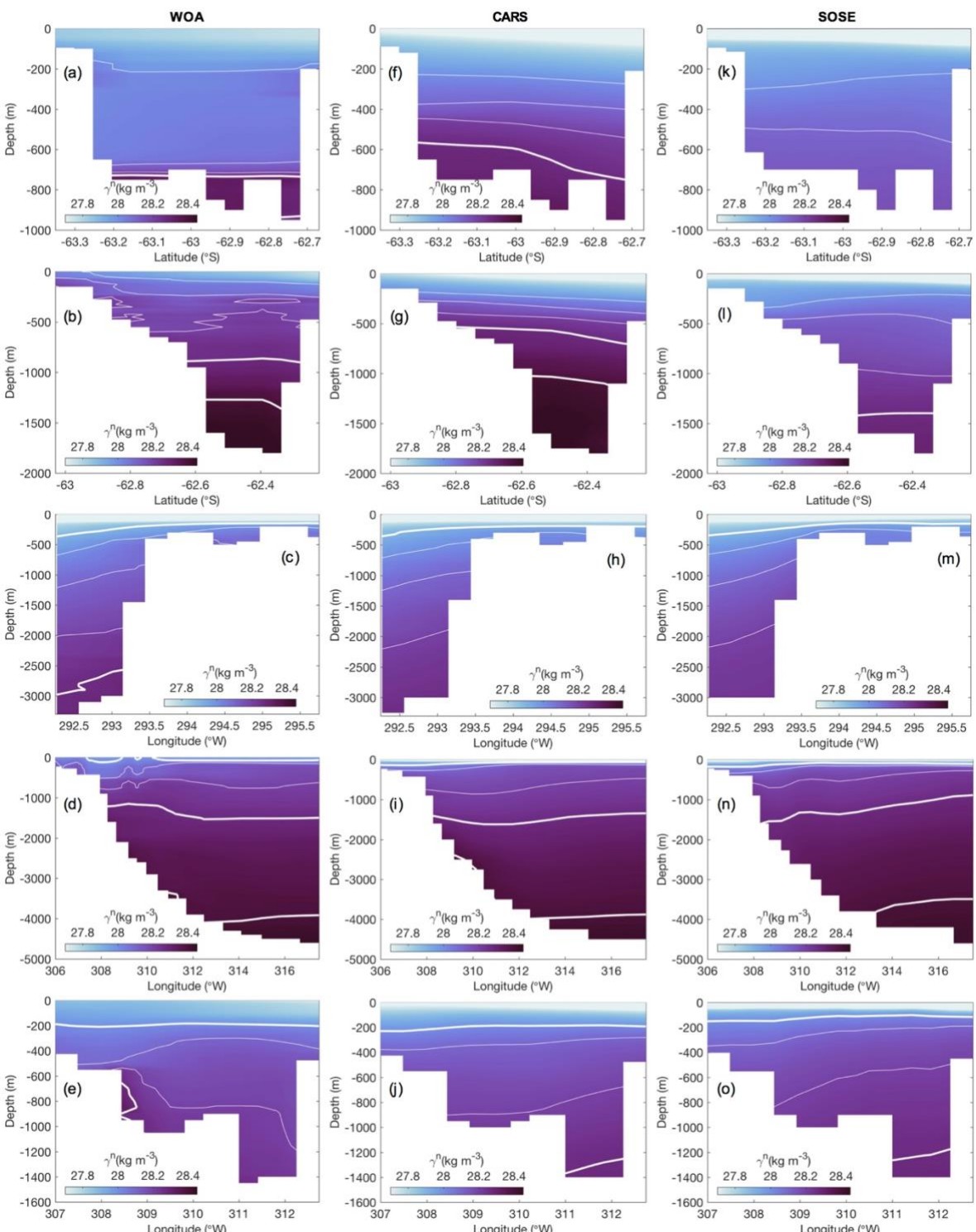

**Figure 15.** Summer vertical sections of neutral density ($\gamma^n$; kg m$^{-3}$) for WOA (left panels), CARS (middle panels) and SOSE (right panels). (a, f, k) Sections between the Livingston Island and the Antarctic Peninsula. Isopycnal of 28.27 kg m$^{-3}$ and 28.4 kg m$^{-3}$ (28 kg m$^{-3}$ to 28.2 kg m$^{-3}$ every 0.1 kg m$^{-3}$) is shown by the thick (thin) white line. (b, g, l) Sections between the King George Island and the Antarctic Peninsula. Isopycnals of 28.27 kg m$^{-3}$ and 28.4 kg m$^{-3}$ (28 kg m$^{-3}$ to 28.2 kg m$^{-3}$ every 0.1 kg m$^{-3}$) is shown by the thick (thin) white line. (c, h, m) Section at the WAP. Isopycnals of 27.9 kg m$^{-3}$ and 28.27 kg m$^{-3}$ (28 kg m$^{-3}$ to 28.2 kg m$^{-3}$ every 0.1 kg m$^{-3}$) is shown by the thick (thin) white line. (d, i, n) Sections WOCE SR4. Isopycnals of 28 kg m$^{-3}$, 28.27 kg m$^{-3}$ and 28.4 kg m$^{-3}$ (28 kg m$^{-3}$ to 28.2 kg m$^{-3}$ every 0.1 kg m$^{-3}$) is shown by the thick (thin) white line. (e, j, o) Section Scotia-Arc. Isopycnals of 28 kg m$^{-3}$ and 28.27 kg m$^{-3}$ (28 kg m$^{-3}$ to 28.2 kg m$^{-3}$ every 0.1 kg m$^{-3}$) is shown by the thick (thin) white line.