# Peer review of "A novel hydrographic gridded data set for the Northern Antarctic Peninsula"

_Earth System Science Data, 2020_

## Referee Comment (RC1) · Anonymous Referee #1 · 4 Nov 2020

This manuscript presents a climatology of the summer conditions in the region of the Northern Antarctic Peninsula, based using cruise data obtained between 2003 and 2019. If the cruise data is undeniably of high value, it is already published and should be freely available. The main novelty here is to propose a gridded product obtained by interpolating optimally all available cruise data, yielding fields of temperature, salinity and dissolved oxygen. Detailed comparisons are then presented between this gridded dataset and standard climatological products available for the same region, showing some minor improvements. The geographical extent of the climatology is however extremely limited, questioning the need for this publication. The authors have chosen to use exclusively cruise data from the Brazilian High Latitude Oceanography Group (GOAL). This restricts highly the value and applicability of this dataset, as the climatology is then mostly limited to the shelf region with very little data along the continental slope surrounding the Northern Antarctic Peninsula. There is yet data available in this region, from Argo, from other cruises or from tagged seals and I assume there should be a lot more available than just what is presented in the NODC comparison in Fig. 9, at least for temperature and salinity. In conclusion, I do not see enough novelty in this publication to warrant publication, as it is not sufficiently exhaustive in the data it incorporates and as it focuses on a region and season that is too limited. I therefore recommend rejection of this manuscript.

---

## Referee Comment (RC2) · Hartmut Hellmer (Referee) · 5 Nov 2020

The submission introduces a gridded hydrographic data set of conservative temperature, absolute salinity, and dissolved oxygen as the mean of austral summer conditions of the period 2003-2019 at the Northern Antarctic Peninsula (NAP). Controlled by complex geography and bottom topography, water masses from Bellingshausen Sea, Weddell Sea, and the Antarctic Circumpolar Current (ACC) meet in Bransfield Strait (BS) creating a unique mixture, which supports high stocks of krill. The latter might be threatened by climate change, which shows one of the strongest signals in Antarctica at the Antarctic Peninsula. The comparison with existing climatologies, based on observations and numerical modelling, unveils some deficiencies of the existing products mainly caused by the lack of spatial resolution. As the resolution of numerical models tend to increase, hydrographic data sets with high-resolution in space and time for model initialization and verification are needed desperately.

Having said this, the presented climatology is an improvement which comes, however, with three caveats:

1. The geographic focus on Bransfield Strait, leaving out – more or less - important regions like the eastern Bellingshausen Sea, southern Drake Passage, northwestern Weddell Sea continental shelf, and Powell Basin.

2. The lack of temporal, at least, seasonal resolution. E.g., only in central BS basin bottom temperatures are close to the surface freezing point (Fig. 12a), indicating some remnants of either local or remote wintertime convection - plus advection.

3. The restriction to GOAL cruises. Knowing that members of the institute participated in two Brazilian comprehensive hydrographic surveys to the northwestern Weddell Sea in 2000 and 2001 onboard of Ary Rongel, I wonder why this data is not included, which would fill the 'data void' to the east (Powell Basin) in the presented climatology.

Numerical models with a regional focus and, thus, high-resolution gain momentum. However, they need to consider the formation area of the water masses flowing towards the area of interest and, for model validation, at least a seasonal resolution. Thus, the presented climatology is of minor value and needs to be extended in space and time. This could be achieved either by combining the GOAL climatology with larger-scale climatologies or to collect more hydrographic data from different sources and platforms to create a comprehensive data set for the continental shelf fringing the whole Antarctic Peninsula.

Nevertheless, I recommend publication of this submission with moderate revision – if possible – because:

1. The GOAL climatology might be of significant value for marine biology, focused on krill and its predators in the NAP area.

2. The GOAL climatology documents the continuous efforts of an institute, which is not a polar institute per se, to build up high standards in education and Antarctic field activities, which are comparable to other polar institutes worldwide.

---

## Author Comment (AC1) · 14 Dec 2020

**Response to the Reviewers**

We are grateful to both Reviewers for their very helpful and constructive feedback. In the following, we outline how we have responded to their comments. Comments by the Reviewers are shown in *black*, and our responses in *blue*.

**Response to Reviewer 1 (Anonymous Referee #1)**

This manuscript presents a climatology of the summer conditions in the region of the Northern Antarctic Peninsula, based using cruise data obtained between 2003 and 2019. If the cruise data is undeniably of high value, it is already published and should be freely available.
Most of the GOAL data sets are already available via PANGAEA website (see below) under moratorium. To access the data, the user needs to fill in a request which is sent to the PIs of the projects. Conversely, the data set of the projects NAUTILUS/INTERBIOTA are still being evaluated by the group (students and postdocs) because they have recently been collected. However, the data set is also available under request to the PIs (mauricio.mata@furg.br; rodrigokerr@furg.br).

List of GOAL data sets available in PANGAEA:

(1) REDE-1
https://doi.pangaea.de/10.1594/PANGAEA.863598 (Mata and Garcia, 2016a)
https://doi.pangaea.de/10.1594/PANGAEA.863599 (Mata and Garcia, 2016b)
https://doi.pangaea.de/10.1594/PANGAEA.863600 (Mata and Garcia, 2016c)

(2) SOS-CLIMATE
https://doi.pangaea.de/10.1594/PANGAEA.864576 (Mata and Garcia, 2016d)
https://doi.pangaea.de/10.1594/PANGAEA.864578 (Mata and Garcia, 2016e)
https://doi.pangaea.de/10.1594/PANGAEA.864579 (Mata and Garcia, 2016f)

(3) POLARCANION/PROASIS
https://doi.pangaea.de/10.1594/PANGAEA.864591 (Mata and Kerr, 2016a)
https://doi.pangaea.de/10.1594/PANGAEA.864592 (Mata and Kerr, 2016b)
https://doi.pangaea.de/10.1594/PANGAEA.864593 (Mata and Kerr, 2016c)

The main novelty here is to propose a gridded product obtained by interpolating optimally all available cruise data, yielding fields of temperature, salinity, and dissolved oxygen. Detailed comparisons are then presented between this gridded dataset and standard climatological products available for the same region, showing some minor improvements.
We respectfully disagree with the reviewer regarding the *minor* improvements. Our gridded product showed major improvements compared to the existing climatologies evaluated (see also comments of Referee #2). For instance, none of the climatologies evaluated successfully represented the characteristic dichotomy in temperature (Figures 16 and A1) and density (Figure A2) of the region as our product did (Figures 13, 14 and 15). The misrepresentation of the regional water masses was associated to saltier (e.g., WOA and CARS) and warmer (e.g., SOSE) waters in the climatologies (Figure 15). Overall, the deep water masses characteristics in the existing climatologies were poorly represented. Representing those waters properly is important because such waters are reminiscent of the shelf water of the Weddell Sea and thus the Bransfield Strait is a proxy region to study changes in those water masses (Dotto et al. 2016, van Caspel et al. 2015, 2018). These flaws in representing highly dynamical regions

could be a result of low-resolution gridding and large interpolation length-scales. These issues are partially overcome in our regional climatology, in which high-resolution gridding and smaller interpolation length-scales are used to better represent the highly-dynamical regions. Finally, as indicated by the Referee #2, "(…) the resolution of numerical models tends to increase, hydrographic data sets with high-resolution in space and time for model initialization and verification are needed desperately." Hence, the development of robust regional climatologies become even more important to attend that demand.

The geographical extent of the climatology is however extremely limited, questioning the need for this publication.
The current climatology is focused on the NAP region. The NAP zone is highly-dynamical in terms of oceanography and it is biologically extremely rich (Kerr et al., 2018a; Atkinson et al., 2020; Costa et al., 2020), being a relevant area that interconnect both the zonal (circumpolar connection of the Southern Ocean) and longitudinal (Subantarctic-Antarctic zones) flows. In addition, the areas around the Antarctic Peninsula have undergone significant atmospheric (Turner et al., 2005) and oceanic warming (Meredith and King, 2005) in the past few decades, which have affected both the water masses (Dotto et al., 2016; Ruiz Barlett et al., 2018) and the biology (Moline et al., 2004; Mendes et al., 2013; Ferreira et al., 2020) of the region, showing its sensitiveness to climate changes.

Regarding the oceanography, the bathymetric configuration of the NAP, mainly the Bransfield Strait, traps considerable amounts of Weddell Sea shelf waters (approximately 60-80% of the deep water mass mixture in the region is formed by LSSW and HSSW, important sources of Antarctic Bottom Water; Gordon et al., 2000; Dotto et al., 2016; Huneke et al., 2016; van Caspel et al., 2018), making it a good proxy region to analyse temporal variability of these shelf waters, giving its easier access and sea ice free conditions in most of the end-spring to early-autumn. The continental shelf of the western and southwestern Weddell Sea historically lacks in situ data due to thick sea ice cover year-round. Thus, the Bransfield Strait is key to analyse those water masses. Moreover, the shelf waters are showing freshening in many parts of the Southern Ocean (Jacobs and Giulivi, 2010; Hellmer et al., 2011; Azaneu et al., 2013; Schmidtko et al., 2014), and its monitoring is important to investigate causes and consequences of those trends. Furthermore, many glaciers at the NAP are still stable (opposite to the southern, West Antarctic Peninsula-WAP region), however if the ocean temperature and/or circulation change, this could lead to higher rates of glacial ice melting (Cook et al., 2016), affecting the regional oceanography and the local food web, with potential impacts on larger scales (Ferreira et al., 2020).

Regarding the biology, the NAP area is an important site for top predators feeding tied to both the abundant krill stock (Atkinson et al., 2020) and highly primary productivity (Costa et al., 2020). If the changes reported for the region continue (e.g., as sea ice cover and seasonality decline, land ice melting, atmospheric and oceanic temperature warming…), it is expected an imbalance or even a collapse in the regional food chain (Ferreira et al., 2020). Moreover, giving its importance, a proposal to make the NAP and the areas around the Antarctic Peninsula a marine protected area is being currently discussed (Hogg et al., 2020).

We also point out to the fact that the Antarctic community has a specific Working Group focusing on the West Antarctic Peninsula and Scotia Arc (WAPSA; http://www.soos.aq/activities/rwg/wapsa) as part of the Southern Ocean Observing System (SOOS). Also, the study region is within the subarea 48.1 of the Commission for the Conservation of Antarctic Marine Living Resources (CCMLAR; https://www.ccamlr.org/en),

which reinforces that a better understanding of the physical, chemical and biological processes is still a challenge in an area that connects different ecosystems under distinct regional and climate-driven influences. In addition, in recent years, different scientific groups published Special Issues about the WAP/NAP environments, highlighting the emerging needs to better comprehension of the region:

1) Polar Biology (2016) - https://link.springer.com/journal/300/volumes-and-issues/39-5;
2) Philosophical Transaction of the Royal Society A (2018) - https://royalsocietypublishing.org/toc/rsta/376/2122;
3) Deep Sea Research II (2019) - https://www.sciencedirect.com/journal/deep-sea-research-part-ii-topical-studies-in-oceanography/vol/149/suppl/C

It is also worth to the mention the contribution of Henley et al. (2019). Variability and change in the west Antarctic Peninsula marine system: Research priorities and opportunities, published recently in Progress in Oceanography (https://doi.org/10.1016/j.pocean.2019.03.003).

Papers cited here (not included in the manuscript references):
- Atkinson et al. 2020 (Nature Climate Change; doi: 10.1038/s41558-018-0370-z)
- Ferreira et al. 2020 (Frontiers in Marine Science; doi: 10.3389/fmars.2020.576254)
- Hogg et al. 2020 (Nature; doi: 10.1038/d41586-020-02939-5)
- Jacobs & Giulivi 2010 (Journal of Climate; doi: 10.1175/2010JCLI3284.1)
- Schmidtko et al. 2014 (Science; doi: 10.1126/science.1256117)
- Turner et al. 2005 (International Journal of Climatology; doi: 10.1002/joc.1130)

The authors have chosen to use exclusively cruise data from the Brazilian High Latitude Oceanography Group (GOAL). This restricts highly the value and applicability of this dataset, as the climatology is then mostly limited to the shelf region with very little data along the continental slope surrounding the Northern Antarctic Peninsula. There is yet data available in this region, from Argo, from other cruises or from tagged seals and I assume there should be a lot more available than just what is presented in the NODC comparison in Fig. 9, at least for temperature and salinity.

Thank you for rising this issue, which was also pointed out by reviewer 2. In this sense, we agree that the final product robustness ought to be significantly improved by the inclusion of additional data sets. Hence, for the revised manuscript, we investigated the data sets available from different platforms to generate a revised product across the NAP, adjacent seas, and transitional zone environments, as suggested by the Referees #1 and #2. Taking into consideration their comments, we combined the GOAL data set with data from NODC (also known as World Ocean Database-WOD) and PANGAEA (both restricted to the period of 1990-2019, to cover the periods of high data standards of the World Ocean Circulation Experiment-WOCE, Climate and Ocean: Variability, Predictability and Change-CLIVAR, and Global Ocean Ship-based Hydrographic Investigations Program-GO-SHIP). We also included ARGO (http://www.coriolis.eu.org/) and Marine Mammal Exploring the Oceans Pole to Pole (MEOP; http://www.meop.net/) data to attend the suggestion from both Referees #1 and #2 regarding the expansion of the temporal representation. Therefore, after the inclusion of those data, we believe that the manuscript has been substantially improved.

Therefore, in attention to the comments from both Referees, we modified our manuscript/product by: (i) expanding our study region to include the areas of the southern Drake Passage, eastern Bellingshausen Sea, northwestern Weddell Sea, Powell Basin and South Orkney Island (Figure R1), and (ii) merging CTD data sets from GOAL, NODC/WOD,

PANGAEA, Hutchinson et al. (2020) at the Larsen C region, ARGO floats and MEOP. These two approaches increased our number of profiles from ~890 (GOAL only) to more than 37000 profiles (considering temperature only; Figure R2). The new expanded study region covers better the transitional oceanographic regimes between the adjacent areas and the NAP, hence, creating an improved climatology to represent in more details the oceanic features. In addition, the inclusion of data from other sources provided basis to create a seasonal climatology, as suggested by the Referee #2. The largest amount of data was provided by MEOP measurements, gathering year-round mainly in shallow coastal areas (Figure R1). The number of profiles generated by the marine mammals can be as high as 18 times, in winter, compared to the other data sources for this region (Figure R2, see temperature). However, most of the MEOP data are restricted to the upper 1000 m due to the physiology and behavior of the animals (Figure R3), and sensors for dissolved oxygen are yet not available. At least ~200 profiles of ARGO floats are provided per season for the study region (Figure R2), including within the Weddell Gyre (Figure R1). These floats measure data up to ~2000 m depth. These caveats have impacts on deeper water masses representations and on the oxygen field, mainly from Autumn to Spring when the coverage of CTD profiles is lesser than Summer. A "caveat" section is included in the revised manuscript to describe clearly the limitations of the new product.

We note that even with the addition of more data, the area of the western Weddell Sea is still poorly sampled (Figure R1). Due to this lack of representation, we masked that region from the climatology. Even though, this new product represented well the study region and it is easy to use for quick investigation and validation of ocean model outputs. An additional feature of our product is the representation of the flow patterns of Weddell Sea shelf waters toward the Bransfield Strait, showing the strait as a proxy region to study temporal changes of those shelf waters properties (Gordon et al., 2000; Dotto et al., 2016; van Caspel et al., 2018).

The validation of the product with CTD measurements has also been made and it is included in the revised manuscript. Comparisons against different climatologies were also made to show the improvement of our high-resolution regional products.

Papers cited here (not included in the manuscript references):
- Hutchinson et al. 2020 (Journal of Geophysical Research; doi: 10.1029/2019JC015855)

[Figure]

Figure R1. New study region covering the area limited to 42˚-68˚W and 59.5˚-66˚S, including the southern Drake Passage, eastern Bellingshausen Sea, northwestern Weddell Sea, Powell Basin and South Orkney Island, and profiles' location included in the new climatology. For this revised version, we merged data from CTD (blue), MEOP (red) and ARGO (green) in different periods of the year (a, Summer; b, Autumn; c, Winter; d, Spring). We note, however, that the western Weddell Sea is still poorly sampled, and high interpolation errors could be found in that region. Consequently, we removed those areas of high errors from the final product. Isobaths of 500, 1000, 3000 and 5000 m are shown by grey solid lines.

[Figure]

Figure R2. Histogram of the number of profiles per season for the different platforms and hydrographic properties for the extended study region shown in Figure R1. Note the different scale in y-axis for the dissolved oxygen. The total number of profiles for each property is given in the title of each panel. The total number of profiles per season is shown in the right upper corner of each panel, and the number of profiles per platform is given in the left upper corner according to CTD (blue), MEOP (red) and ARGO (green).

[Figure]

Figure R3. Number of samples per depth. First row represents in situ temperature measurements, middle row is for practical salinity and bottom row is for dissolved oxygen. The first column (from the left to the right) shows the profiles considering all platforms, the second column is for CTD, the third column for ARGO and the fourth column for MEOP. Note the different x-axis scales between platforms and for the dissolved oxygen. The line colours represent the seasons: Summer (red), Autumn (cyan), Winter (blue) and Spring (green).

In conclusion, I do not see enough novelty in this publication to warrant publication, as it is not sufficiently exhaustive in the data it incorporates and as it focuses on a region and season that is too limited. I therefore recommend rejection of this manuscript.

We thank again the Referee #1 for taking time to review our manuscript and for the many suggestions. We accepted his/her/their suggestions, and we improved the amount of data, expanded the study region and incorporated a seasonal varying climatology for this important and unique area of the Southern Ocean, which can provide insights on the temporal variability of the sub-optimally sampled Weddell Sea shelf waters. We hope that our responses and efforts to improve this manuscript are enough to change the Reviewer's opinion and consider this work worth of publishing.

**Response to Reviewer 2 (Dr. Hartmut Hellmer, hartmut.hellmer@awi.de)**

The submission introduces a gridded hydrographic data set of conservative temperature, absolute salinity, and dissolved oxygen as the mean of austral summer conditions of the period 2003-2019 at the Northern Antarctic Peninsula (NAP). Controlled by complex geography and bottom topography, water masses from Bellingshausen Sea, Weddell Sea, and the Antarctic Circumpolar Current (ACC) meet in Bransfield Strait (BS) creating a unique mixture, which supports high stocks of krill. The latter might be threatened by climate change, which shows one of the strongest signals in Antarctica at the Antarctic Peninsula. The comparison with existing climatologies, based on observations and numerical modelling, unveils some deficiencies of the existing products mainly caused by the lack of spatial resolution. As the resolution of numerical models tend to increase, hydrographic data sets with high-resolution in space and time for model initialization and verification are needed desperately.

We thank Dr. Hartmut Hellmer for taking time to perform this detailed review and for his positive feedback about our work. We followed his suggestions and improved our product by: (i) combining different data sets such as NODC/WOD, PANGAEA, GOAL and Hutchinson et al. (2020; measurements collected off Larsen C Ice Shelf) to improve the representation of the study region based on a larger period (i.e., 1990 to 2019) and more areas; (ii) evaluating the inclusion of MEOP and ARGO data sets to adjust the climatology to show temporal resolution as well; and, (iii) extending the representation of the climatology to areas of eastern Bellingshausen Sea, southern Drake Passage, northwestern Weddell Sea continental shelf and Powell Basin. All considerations are discussed below.

Having said this, the presented climatology is an improvement which comes, however, with three caveats:

1. The geographic focus on Bransfield Strait, leaving out – more or less - important regions like the eastern Bellingshausen Sea, southern Drake Passage, northwestern Weddell Sea continental shelf, and Powell Basin.

The initial focus was the Bransfield and Gerlache straits because these are the main regions of the NAP and have been extensively covered by the GOAL group. However, by adding the new data sets as mentioned above, we have more information to represent the transitional areas between the adjacent regions cited by the Referee and the NAP. Thus, these important regions are now included in the new version of the product/manuscript (see Figure R1). We expanded the product to cover the coordinates 42˚-68˚W and 59.5˚-66˚S. The grid spacing (~10 km) was kept the same and the vertical levels were reduced to 90 levels between 5 m depth to 6000 m depth, with spacing increasing from 5 m in the top 50 m depth to 500 m below 4000 m depth. Given the spatial limitation of CTD measurements compared to ARGO and MEOP (Figure R3), the product has now considerably less data for deeper depths than 2000 m for the Weddell Sea, Bellingshausen Sea and Drake Passage. A section describing the caveats of the new product is included in the revised manuscript.

2. The lack of temporal, at least, seasonal resolution. E.g., only in central BS basin bottom temperatures are close to the surface freezing point (Fig. 12a), indicating some remnants of either local or remote wintertime convection - plus advection.

The temporal representation in the Southern Ocean is now possible with the inclusion of ARGO and MEOP data sets. However, we point out to the fact that although the amount of ARGO data is relatively high offshore of the shallow bays (i.e., from the open ocean to the continental slope; Figure R1), the profiles are depth-limited to 2000 m (Figure R3). Similarly, the amount of MEOP data is relatively high on coastal waters of the NAP and eastern Bellingshausen Sea (Figure R1), but they too have a depth limitation between ~500 m to ~1500 m (Figure R3).

Having said that, we included both ARGO and MEOP data sets to create a seasonal climatology, at least, for the upper ocean. Most of the deep profiles are possible only by CTD casts (Figure R3), which are reduced in all seasons when compared to summer (Figure R2). With the development of deep ARGO, one expects a better representation of the deep basins. Therefore, we created a seasonal climatology (i.e., Summer, Autumn, Winter and Spring) to cover more regions and we also included more data to have a higher reliability of the study region. Given the limitations of dissolved oxygen data (mostly collected by CTD and currently by ARGO), this property has less coverage than temperature and salinity. This is also a caveat, and it is discussed in the revised manuscript.

3. The restriction to GOAL cruises. Knowing that members of the institute participated in two Brazilian comprehensive hydrographic surveys to the northwestern Weddell Sea in 2000 and 2001 onboard of Ary Rongel, I wonder why this data is not included, which would fill the 'data void' to the east (Powell Basin) in the presented climatology.

The data mentioned by the Referee are now included in the new product, which also counts with many historical data from different sources to reduce these areas of 'data void'. Previously, we were working with ~890 hydrographic station from GOAL collected between 2003-2019. Now, after the inclusion of the different platforms, increasing the time span (1990-2019) and expanding the study area, our data set increased up to ~37000 profiles (Figure R2). With this richer data set, we can cover more areas than before, and we could attend the referee suggestions regarding the inclusion of more data/platforms, expansion of the study region and expansion of temporal representation.

Numerical models with a regional focus and, thus, high-resolution gain momentum. However, they need to consider the formation area of the water masses flowing towards the area of interest and, for model validation, at least a seasonal resolution. Thus, the presented climatology is of minor value and needs to be extended in space and time. This could be achieved either by combining the GOAL climatology with larger-scale climatologies or to collect more hydrographic data from different sources and platforms to create a comprehensive data set for the continental shelf fringing the whole Antarctic Peninsula.

We thank Dr. Hellmer for his suggestion. We expanded the study area to cover more regions adjacent to the Antarctic Peninsula (Figure R1). However, some dense water mass formation areas, such as the western Weddell Sea, are historically undersampled due to the thick sea ice cover most of the year. Even marine mammals seem to avoid those areas (Figure R1). Thus, the interpolation in this region created larger errors, not useful to validate the ocean models. Because of that, we masked these areas in our regional climatology. The extension of the study region, however, highlights even better now the flow tracks of dense shelf waters toward the Bransfield Strait in the different seasons.

Nevertheless, I recommend publication of this submission with moderate revision – if possible – because:
1. The GOAL climatology might be of significant value for marine biology, focused on krill and its predators in the NAP area.
2. The GOAL climatology documents the continuous efforts of an institute, which is not a polar institute per se, to build up high standards in education and Antarctic field activities, which are comparable to other polar institutes worldwide.

We are grateful to Dr. Hellmer for considering the manuscript worth of publication after some moderate review. We are especially grateful to him for highlighting the efforts of a relatively small group, which has been working tirelessly in the region for the last 20 years, even considering all possible logistical, financial and human resources barriers. The regions

surrounding the Antarctic Peninsula are biologically rich and vulnerable to climate changes (e.g. Kerr et al., 2018a; Ferreira et al., 2020). Thus, having a NAP hydrographic gridded product available will be helpful to the scientific community working in those areas. We believe that the new version with more data from different sources has expanded the study region and presents now a better temporal representation.

---

## Author Response (AR1)

**Summary of changes**

The major comments given by the reviewers were regarding the size of study region and amount of data set used. In this new version of the manuscript, we have addressed these questions by:

- Expanding the study region to cover not only the NAP region, but also the transitional and adjacent regions. The new region has limits of 59.5°-66°S and 42°-68°W (Line 130), and include the Bransfield and Gerlache straits, the easternmost Bellingshausen Sea, southern Drake Passage, Powell Basin, Scotia Arc and northwestern Weddell Sea (Figures 1 and 2; Lines 119-130).
  - 2) Merging CTD data from the World Ocean Database, Pangaea, Hutchinson et al. (2020) off Larsen region and GOAL with Argo data at offshore regions and MEOP data set mostly at the West Antarctic Peninsula (WAP) and NAP. The data was restricted to the period of 1990-2019 to cover the high standards period of WOCE, CLIVAR and GO-SHIP (Lines 119-130);
    - 3) Creating a seasonal climatology with outputs for summer (Jan-Mar), autumn (Apr-Jun), winter (Jul-Sep) and spring (Oct-Dec) (Lines 131-131).
- 15 Section 2 (Hydrographic data and ancillary data set) describes the new study area and brings information on the new data sets used, such as temporal resolution, spatial and vertical coverage (Figures 2-3 and S1-S2 and Lines 137-152) and accuracy of the different platforms (Lines 153-162). Because of the inclusion of the new data sets, we had to change a few settings of the interpolation (smoothing) scheme (Lines 201-202) and reduce the depth levels to 90 (new Table 1). These changes are now described in Section 3 (Methods).
- 20

10

All figures were recreated based on these changes incorporated in the new version of the manuscript. Moreover, many parts of the text were reformulated to include the new features observed in the climatology, called now NAPv1.0. In total, we have 15 new figures included in the main text and 10 new figures included as supplementary material. Section 4.1 (*Reasonableness check of the gridded product*) compares the NAPv1.0 outputs against CTD measurements to check the robustness of the

- 25 interpolation and the product. The evaluation of the gridded product against independent data sets was removed to avoid comparing product with older CTD cast (<1990s), which accuracy is not as high as the standards developed during the 1990s. Section 4.2 (*Representation of the main hydrographic features in the NAP*) describes the main results of the climatology on seasonal time scales for horizontal maps of property distribution and vertical sections. Due to the new expanded area, now we have also tested the representation of the NAPv1.0 in sections out of the Bransfield Strait (Figures 11, 14 and 15). In this new
- 30 version of the manuscript, we have also compared our climatology against other climatologies (Section 4.3. Comparison against other climatological products) for the sections within the NAP and out of the Bransfield Strait. A new section (Section 4.4 Caveats of the NAPv1.0 climatology, Lines 411-440) has been included to discuss a few caveats of the climatology, such as spatial, vertical and temporal limitations of the input data, which has limited the representativeness of the climatology in some seasons.

The inclusion of the new data sets has improved both the climatology representation and the inflows towards the NAP, providing new knowledge on a larger context. However, the main conclusion has not changed: regional climatologies are needed to better resolve scales not represented by large-scale climatologies. Thus, "the NAPv1.0 is a valuable tool for setting up regional ocean models and ocean reanalysis assessment, or to any user who wants to use it to characterize the mean-state hydrography of the NAP in the end of 20th-century and early 21st-century".

**Response to the Reviewers (point-by-point responses)**

We are grateful to both Reviewers for their very helpful and constructive feedback. Below, we provide point-by-point responses to the reviews including a list of all relevant changes made in the manuscript. Comments by the Reviewers are shown in *black*, our responses in *blue* (note that they are the same of the open discussion file), and the new responses showing the modification in the manuscript are in *red*. The track changes file is attached below the responses, where the modified parts of the new manuscript are in *blue*.

**50 **Response to Reviewer 1 (Anonymous Referee #1)**

This manuscript presents a climatology of the summer conditions in the region of the Northern Antarctic Peninsula, based using cruise data obtained between 2003 and 2019. If the cruise data is undeniably of high value, it is already published and should be freely available.

- 55 Most of the GOAL data sets are already available via PANGAEA website (see below) under moratorium. To access the data, the user needs to fill in a request which is sent to the PIs of the projects. Conversely, the data set of the projects NAUTILUS/INTERBIOTA are still being evaluated by the group (students and postdocs) because they have recently been collected. However, the data set is also available under request to the PIs (mauricio.mata@furg.br; rodrigokerr@furg.br).
- 60 List of GOAL data sets available in PANGAEA:

**(1) **REDE-1**

65

https://doi.pangaea.de/10.1594/PANGAEA.863598 (Mata and Garcia, 2016a) https://doi.pangaea.de/10.1594/PANGAEA.863599 (Mata and Garcia, 2016b) https://doi.pangaea.de/10.1594/PANGAEA.863600 (Mata and Garcia, 2016c)

**(2) SOS-CLIMATE**

https://doi.pangaea.de/10.1594/PANGAEA.864576 (Mata and Garcia, 2016d)

35

https://doi.pangaea.de/10.1594/PANGAEA.864578 (Mata and Garcia, 2016e)

70 https://doi.pangaea.de/10.1594/PANGAEA.864579 (Mata and Garcia, 2016f)

**(3) POLARCANION/PROASIS**

https://doi.pangaea.de/10.1594/PANGAEA.864591 (Mata and Kerr, 2016a) https://doi.pangaea.de/10.1594/PANGAEA.864592 (Mata and Kerr, 2016b)

75 https://doi.pangaea.de/10.1594/PANGAEA.864593 (Mata and Kerr, 2016c)

The main novelty here is to propose a gridded product obtained by interpolating optimally all available cruise data, yielding fields of temperature, salinity, and dissolved oxygen. Detailed comparisons are then presented between this gridded dataset and standard climatological products available for the same region, showing some minor improvements.

- 80 We respectfully disagree with the reviewer regarding the *minor* improvements. Our gridded product showed major improvements compared to the existing climatologies evaluated (see also comments of Referee #2). For instance, none of the climatologies evaluated successfully represented the characteristic dichotomy in temperature (Figures 16 and A1 from the older manuscript) and density (Figure A2 from the older manuscript) of the region as our product did (Figures 13, 14 and 15 from the older manuscript). The misrepresentation of the regional water masses was associated to saltier (e.g., WOA and
- 85 CARS) and warmer (e.g., SOSE) waters in the climatologies (Figure 15 from the older manuscript). Overall, the deep water masses characteristics in the existing climatologies were poorly represented. Representing those waters properly is important because such waters are reminiscent of the shelf water of the Weddell Sea and thus the Bransfield Strait is a proxy region to study changes in those water masses (Dotto et al. 2016, van Caspel et al. 2015, 2018). These flaws in representing highly dynamical regions could be a result of low-resolution gridding and large interpolation length-scales. These issues are partially
- 90 overcome in our regional climatology, in which high-resolution gridding and smaller interpolation length-scales are used to better represent the highly-dynamical regions. Finally, as indicated by the Referee #2, "(...) the resolution of numerical models tends to increase, hydrographic data sets with high-resolution in space and time for model initialization and verification are needed desperately." Hence, the development of robust regional climatologies become even more important to attend that demand.
- 95 In regards of the reviewer's comments, we can list several improvements seen in our climatology: (i) better representation of the inflows of Weddell Sea-sourced waters and the inflow of warm waters from the Bellingshausen Sea, (ii) the Peninsula and the Bransfield Fronts, (iii) the cyclonic circulation within the Bransfield Strait, (iv) the Bransfield Strait central and eastern basins dichotomy hydrographic regime, (v) the narrow flow of cold and oxygen-rich water along the continental slope of the Weddell Sea inflowing toward the Bransfield Strait, and (vi) the downslope flow of recently ventilated shelf waters at the
- 100 Weddell Sea continental slope (also observed in CARS climatology). Besides, our climatology has better representation of the water masses within the central NAP, showing the improvements of focusing on regional products.

- 105 The geographical extent of the climatology is however extremely limited, questioning the need for this publication.
- The current climatology is focused on the NAP region. The NAP zone is highly-dynamical in terms of oceanography and it is biologically extremely rich (Kerr et al., 2018a; Atkinson et al., 2020; Costa et al., 2020), being a relevant area that interconnect both the zonal (circumpolar connection of the Southern Ocean) and longitudinal (Subantarctic-Antarctic zones) flows. In addition, the areas around the Antarctic Peninsula have undergone significant atmospheric (Turner et al., 2005) and oceanic
- 110 warming (Meredith and King, 2005) in the past few decades, which have affected both the water masses (Dotto et al., 2016; Ruiz Barlett et al., 2018) and the biology (Moline et al., 2004; Mendes et al., 2013; Ferreira et al., 2020) of the region, showing its sensitiveness to climate changes.

Regarding the oceanography, the bathymetric configuration of the NAP, mainly the Bransfield Strait, traps considerable amounts of Weddell Sea shelf waters (approximately 60-80% of the deep water mass mixture in the region is formed by LSSW and HSSW, important sources of Antarctic Bottom Water; Gordon et al., 2000; Dotto et al., 2016; Huneke et al., 2016; van Caspel et al., 2018), making it a good proxy region to analyse temporal variability of these shelf waters, giving its easier access and sea ice free conditions in most of the end-spring to early-autumn. The continental shelf of the western and southwestern Weddell Sea historically lacks in situ data due to thick sea ice cover year-round. Thus, the Bransfield Strait is key to analyse

- 120 those water masses. Moreover, the shelf waters are showing freshening in many parts of the Southern Ocean (Jacobs and Giulivi, 2010; Hellmer et al., 2011; Azaneu et al., 2013; Schmidtko et al., 2014), and its monitoring is important to investigate causes and consequences of those trends. Furthermore, many glaciers at the NAP are still stable (opposite to the southern, West Antarctic Peninsula-WAP region), however if the ocean temperature and/or circulation change, this could lead to higher rates of glacial ice melting (Cook et al., 2016), affecting the regional oceanography and the local food web, with potential
- 125 impacts on larger scales (Ferreira et al., 2020).

Regarding the biology, the NAP area is an important site for top predators feeding tied to both the abundant krill stock (Atkinson et al., 2020) and highly primary productivity (Costa et al., 2020). If the changes reported for the region continue (e.g., as sea ice cover and seasonality decline, land ice melting, atmospheric and oceanic temperature warming...), it is
expected an imbalance or even a collapse in the regional food chain (Ferreira et al., 2020). Moreover, giving its importance, a proposal to make the NAP and the areas around the Antarctic Peninsula a marine protected area is being currently discussed

(Hogg et al., 2020).

We also point out to the fact that the Antarctic community has a specific Working Group focusing on the West Antarctic

135 Peninsula and Scotia Arc (WAPSA; http://www.soos.aq/activities/rwg/wapsa) as part of the Southern Ocean Observing System (SOOS). Also, the study region is within the subarea 48.1 of the Commission for the Conservation of Antarctic Marine

Living Resources (CCMLAR; https://www.ccamlr.org/en), which reinforces that a better understanding of the physical, chemical and biological processes is still a challenge in an area that connects different ecosystems under distinct regional and climate-driven influences. In addition, in recent years, different scientific groups published Special Issues about the WAP/NAP environments, highlighting the emerging needs to better comprehension of the region:

- 1) Polar Biology (2016) https://link.springer.com/journal/300/volumes-and-issues/39-5;
- 2) Philosophical Transaction of the Royal Society A (2018) https://royalsocietypublishing.org/toc/rsta/376/2122;
- 3) Deep Sea Research II (2019) https://www.sciencedirect.com/journal/deep-sea-research-part-ii-topical-studies-in-
- 145 oceanography/vol/149/suppl/C

It is also worth to the mention the contribution of Henley et al. (2019). Variability and change in the west Antarctic Peninsula marine system: Research priorities and opportunities, published recently in Progress in Oceanography (https://doi.org/10.1016/j.pocean.2019.03.003).

150

140

- Papers cited here (not included in the manuscript references):
- Atkinson et al. 2020 (Nature Climate Change; doi: 10.1038/s41558-018-0370-z)
- Ferreira et al. 2020 (Frontiers in Marine Science; doi: 10.3389/fmars.2020.576254)
- Hogg et al. 2020 (Nature; doi: 10.1038/d41586-020-02939-5)
- 155 Jacobs & Giulivi 2010 (Journal of Climate; doi: 10.1175/2010JCLI3284.1)
  - Schmidtko et al. 2014 (Science; doi: 10.1126/science.1256117)
  - Turner et al. 2005 (International Journal of Climatology; doi: 10.1002/joc.1130)

We have expanded the study region as suggested by the reviewers. The full response to this issue is addressed in the next question.

The authors have chosen to use exclusively cruise data from the Brazilian High Latitude Oceanography Group (GOAL). This restricts highly the value and applicability of this dataset, as the climatology is then mostly limited to the shelf region with very little data along the continental slope surrounding the Northern Antarctic Peninsula. There is yet data available in this

165 region, from Argo, from other cruises or from tagged seals and I assume there should be a lot more available than just what is presented in the NODC comparison in Fig. 9, at least for temperature and salinity.

Thank you for rising this issue, which was also pointed out by reviewer 2. In this sense, we agree that the final product robustness ought to be significantly improved by the inclusion of additional data sets. Hence, for the revised manuscript, we investigated the data sets available from different platforms to generate a revised product across the NAP, adjacent seas, and

170 transitional zone environments, as suggested by the Referees #1 and #2. Taking into consideration their comments, we combined the GOAL data set with data from NODC (also known as World Ocean Database-WOD) and PANGAEA (both

restricted to the period of 1990-2019, to cover the periods of high data standards of the World Ocean Circulation Experiment-WOCE, Climate and Ocean: Variability, Predictability and Change-CLIVAR, and Global Ocean Ship-based Hydrographic Investigations Program-GO-SHIP). We also included Argo (http://www.coriolis.eu.org/) and Marine Mammal Exploring the

- 175 Oceans Pole to Pole (MEOP; http://www.meop.net/) data to attend the suggestion from both Referees #1 and #2 regarding the expansion of the temporal representation. Therefore, after the inclusion of those data, we believe that the manuscript has been substantially improved.
- Therefore, in attention to the comments from both Referees, we modified our manuscript/product by: (i) expanding our study
  region to include the areas of the southern Drake Passage, eastern Bellingshausen Sea, northwestern Weddell Sea, Powell Basin and South Orkney Island (new Figures 1 and 2), and (ii) merging CTD data sets from GOAL, NODC/WOD, PANGAEA, Hutchinson et al. (2020) at the Larsen C region, Argo floats and MEOP (Lines 120-162). These two approaches increased our number of profiles from ~890 (GOAL only) to more than 37000 profiles (considering temperature only; new Figure 3). The new expanded study region covers better the transitional oceanographic regimes between the adjacent areas and the NAP, hence, creating an improved climatology to represent in more details the oceanic features. In addition, the inclusion of data from other sources provided basis to create a seasonal climatology (new Figures 2, 6, 7, 8 and 9), as suggested by the Referee
- #2. The largest amount of data was provided by MEOP measurements, gathering year-round mainly in shallow coastal areas (Figure R1). The number of profiles generated by the marine mammals can be as high as 18 times, in winter, compared to the other data sources for this region (new Figure 3, see temperature). However, most of the MEOP data are restricted to the upper
- 190 1000 m due to the physiology and behaviour of the animals (new Figures 3 and S3), and sensors for dissolved oxygen are yet not available. At least ~200 profiles of Argo floats are provided per season for the study region (new Figure 3), including within the Weddell Gyre (new Figure 2). These floats measure data up to ~2000 m depth (new Figures 3 and S3). These caveats have impacts on deeper water masses representations and on the oxygen field, mainly from autumn to spring when the coverage of CTD profiles is lesser than summer. A "caveat" section is included in the revised manuscript to describe clearly the limitations of the new product.
  - We note that even with the addition of more data, the area of the western Weddell Sea is still poorly sampled (new Figure 2). Due to this lack of representation, we masked that region from the climatology. Even though, this new product represented well the study region and it is easy to use for quick investigation and validation of ocean model outputs. An additional feature
- 200 of our product is the representation of the flow patterns of Weddell Sea shelf waters toward the Bransfield Strait, showing the strait as a proxy region to study temporal changes of those shelf waters properties (Gordon et al., 2000; Dotto et al., 2016; van Caspel et al., 2018).

The whole section 4.2 (*Representation of the main hydrographic features in the NAP*) has been modified to include the new observations of the expanded climatology in summer and other seasons. Lines 278-283, 302-309 and 327-330 brings

discussions regarding the seasonal representation of the new outputs. A new section 4.4 (*Caveats of the NAPv1.0 climatology*) has been included to discuss the caveats of the climatology, such as limitation of vertical representation in seasons other than summer given the scarceness of CTD measurements.

210 The validation of the product with CTD measurements has also been made and it is included in the revised manuscript. Comparisons against different climatologies were also made to show the improvement of our high-resolution regional products.

Section 4.1 (*Reasonableness check of the gridded product*) has been modified to include the new comparison of the climatology with CTD data only to show the robustness of the interpolation and the representation of the magnitude of the hydrographic data in the study region (Lines 218-235 and Figures 5 and S3).

Papers cited here (not included in the manuscript references):

- Hutchinson et al. 2020 (Journal of Geophysical Research; doi: 10.1029/2019JC015855)

220 In conclusion, I do not see enough novelty in this publication to warrant publication, as it is not sufficiently exhaustive in the data it incorporates and as it focuses on a region and season that is too limited. I therefore recommend rejection of this manuscript.

We thank again the Referee #1 for taking time to review our manuscript and for the many suggestions. We accepted his/her/their suggestions, and we improved the amount of data, expanded the study region and incorporated a seasonal varying

225 climatology for this important and unique area of the Southern Ocean, which can provide insights on the temporal variability of the sub-optimally sampled Weddell Sea shelf waters. We hope that our responses and efforts to improve this manuscript are enough to change the Reviewer's opinion and consider this work worth of publishing.

230

The submission introduces a gridded hydrographic data set of conservative temperature, absolute salinity, and dissolved oxygen as the mean of austral summer conditions of the period 2003-2019 at the Northern Antarctic Peninsula (NAP).

- 235 Controlled by complex geography and bottom topography, water masses from Bellingshausen Sea, Weddell Sea, and the Antarctic Circumpolar Current (ACC) meet in Bransfield Strait (BS) creating a unique mixture, which supports high stocks of krill. The latter might be threatened by climate change, which shows one of the strongest signals in Antarctica at the Antarctic Peninsula. The comparison with existing climatologies, based on observations and numerical modelling, unveils some deficiencies of the existing products mainly caused by the lack of spatial resolution. As the resolution of numerical models
- 240 tend to increase, hydrographic data sets with high-resolution in space and time for model initialization and verification are needed desperately.

We thank Dr. Hartmut Hellmer for taking time to perform this detailed review and for his positive feedback about our work. We followed his suggestions and improved our product by: (i) combining different data sets such as NODC/WOD, PANGAEA, GOAL and Hutchinson et al. (2020; measurements collected off Larsen C Ice Shelf) to improve the representation

- of the study region based on a larger period (i.e., 1990 to 2019) and more areas; (ii) evaluating the inclusion of MEOP and Argo data sets to adjust the climatology to show temporal resolution as well; and, (iii) extending the representation of the climatology to areas of eastern Bellingshausen Sea, southern Drake Passage, northwestern Weddell Sea continental shelf and Powell Basin. All considerations are discussed below.
- 250 Having said this, the presented climatology is an improvement which comes, however, with three caveats:

 The geographic focus on Bransfield Strait, leaving out – more or less - important regions like the eastern Bellingshausen Sea, southern Drake Passage, northwestern Weddell Sea continental shelf, and Powell Basin.

The initial focus was the Bransfield and Gerlache straits because these are the main regions of the NAP and have been extensively covered by the GOAL group. However, by adding the new data sets as mentioned above, we have more information

- 255 to represent the transitional areas between the adjacent regions cited by the Referee and the NAP. Thus, these important regions are now included in the new version of the product/manuscript (new Figure 1-2). We expanded the product to cover the coordinates 42°-68°W and 59.5°-66°S. The grid spacing (~10 km) was kept the same and the vertical levels were reduced to 90 levels between 5 m depth to 6000 m depth, with spacing increasing from 5 m in the top 50 m depth to 500 m below 4000 m depth (new Table 1). Given the spatial limitation of CTD measurements compared to Argo and MEOP (new Figure 3), the
- 260 product has now considerably less data for deeper depths than 2000 m for the Weddell Sea, Bellingshausen Sea and Drake Passage. A section describing the caveats of the new product is included in the revised manuscript (Section 4.4 Caveats of the NAPv1.0 climatology).

2. The lack of temporal, at least, seasonal resolution. E.g., only in central BS basin bottom temperatures are close to the surface

- 265 freezing point (Fig. 12a), indicating some remnants of either local or remote wintertime convection plus advection. The temporal representation in the Southern Ocean is now possible with the inclusion of Argo and MEOP data sets. However, we point out to the fact that although the amount of Argo data is relatively high offshore of the shallow bays (i.e., from the open ocean to the continental slope; new Figure 2), the profiles are depth-limited to 2000 m (new Figure 3 and S2). Similarly, the amount of MEOP data is relatively high on coastal waters of the NAP and eastern Bellingshausen Sea (new Figure 2), but
- 270 they too have a depth limitation between ~500 m to ~1500 m (new Figures 3 and S2). Having said that, we included both Argo and MEOP data sets to create a seasonal climatology, at least, for the upper ocean. Most of the deep profiles are possible only by CTD casts (new Figure 3 and S2), which are reduced in all seasons when compared to summer (new Figure 3). With the development of deep Argo, one expects a better representation of the deep basins. Therefore, we created a seasonal climatology (i.e., summer, autumn, winter and spring) to cover more regions and we also included more data to have a higher reliability of
- 275 the study region. Given the limitations of dissolved oxygen data (mostly collected by CTD and currently by Argo), this property has less coverage than temperature and salinity. This is also a caveat, and it is discussed in the revised manuscript (new Section 4.2).

In Section 4.2, we discuss the new observations of the new product including on seasonal time-scales (Lines 278-283, 302-309 and 327-330). Figures 6, 7, 8, 9 and Supplementary Figures S4-S9 show the representation of the climatology for all seasons. The density of data is higher in summer and thus the representativeness is closer to reality. Therefore, we focused on addressing most of our comparisons for that season.

The restriction to GOAL cruises. Knowing that members of the institute participated in two Brazilian comprehensive
 hydrographic surveys to the northwestern Weddell Sea in 2000 and 2001 onboard of Ary Rongel, I wonder why this data is not included, which would fill the 'data void' to the east (Powell Basin) in the presented climatology.

The data mentioned by the Referee are now included in the new product, which also counts with many historical data from different sources to reduce these areas of 'data void'. Previously, we were working with ~890 hydrographic station from GOAL collected between 2003-2019. Now, after the inclusion of the different platforms, increasing the time span (1990-2019) and expanding the study area, our data set increased up to ~37000 profiles (new Figure 3). With this richer data set, we can cover

more areas than before, and we could attend the referee suggestions regarding the inclusion of more data/platforms, expansion of the study region and expansion of temporal representation.

The new study region and the representativeness of the regional oceanographic features after the inclusion of more data sets as recommended by the reviewers can be observed in Figure 6 to 9. We also discuss the new improvements in section 4.2 (*Representation of the main hydrographic features in the NAP*). Numerical models with a regional focus and, thus, high-resolution gain momentum. However, they need to consider the formation area of the water masses flowing towards the area of interest and, for model validation, at least a seasonal resolution.

- 300 Thus, the presented climatology is of minor value and needs to be extended in space and time. This could be achieved either by combining the GOAL climatology with larger-scale climatologies or to collect more hydrographic data from different sources and platforms to create a comprehensive data set for the continental shelf fringing the whole Antarctic Peninsula. We thank Dr. Hellmer for his suggestion. We expanded the study area to cover more regions adjacent to the Antarctic Peninsula
- (new Figures 1 and 2). However, some dense water mass formation areas, such as the western Weddell Sea, are historically
  undersampled due to the thick sea ice cover most of the year. Even marine mammals seem to avoid those areas (new Figure 2). Thus, the interpolation in this region created larger errors, not useful to validate the ocean models. Because of that, we masked these areas in our regional climatology (Lines 184-187). The extension of the study region, however, highlights even better now the flow tracks of dense shelf waters toward the Bransfield Strait in the different seasons. For instance, a cold and oxygen-rich flow following the continental slope of the Weddell Sea and entering the Bransfield Strait is observed in this new
  product (Figure 6), as well as the downslope flow of new dense water at the slope of the Section SR4 in the northwestern
- 310 product (Figure 6), as well as the downslope flow of new dense water at the slope of the Section SR4 in the northwestern Weddell Sea (Figure 11).

Nevertheless, I recommend publication of this submission with moderate revision - if possible - because:

1. The GOAL climatology might be of significant value for marine biology, focused on krill and its predators in the NAP area.

- 2. The GOAL climatology documents the continuous efforts of an institute, which is not a polar institute per se, to build up high standards in education and Antarctic field activities, which are comparable to other polar institutes worldwide. We are grateful to Dr. Hellmer for considering the manuscript worth of publication after some moderate review. We are especially grateful to him for highlighting the efforts of a relatively small group, which has been working tirelessly in the region for the last 20 years, even considering all possible logistical, financial and human resources barriers. The regions
- 320 surrounding the Antarctic Peninsula are biologically rich and vulnerable to climate changes (e.g. Kerr et al., 2018a; Ferreira et al., 2020). Thus, having a NAP hydrographic gridded product available will be helpful to the scientific community working in those areas. We believe that the new version with more data from different sources has expanded the study region and presents now a better temporal representation.

[revised manuscript text omitted]

---

## Author Response (AR2)

**Response to the Topical Editor**

Dear Editor,

5    We are grateful to the topical Editor for his positive evaluation and decision regarding our manuscript. Below, we provide the point-by-point response to the topical editor's comment and the list of relevant changes including the lines and figure number. The editor's comment is shown in *black*, our responses are shown in *blue*. The track changes file is attached below the responses, where the modified parts are shown in *red*.

10   line 22: datasets must be available in repository assuring the long term preservation and a persistent identifier. "goal.furg.br" is not having these two characteristics. Please delete this link.

We removed the link from the abstract.

Line 153: please delete this part of the sentence "given their distinct frequency sampling"

15   Done.

The figures are many and make reading of the text slow. I recommend moving figures 6, 7, 8, 9 in the supplement. In the text you could condense the four figures into one by showing the surface maps as an example and referring to the supplement for more details. Of course, the text must be changed accordingly.

20   Thank you for the suggestion. We merged figures 6, 7, 8 and 9 into 2 figures (new Figs. 6 and 7). Figure 6 shows the seasonal representation of conservative temperature and absolute salinity at 10 m depth. Figure 7 shows the seasonal representation of neutral density and dissolved oxygen at 10 m depth. We decided to merge the figures into 2 figures (instead of merging into only one, as suggested) to have panels with a good size and resolution for the final version of the manuscript. The new supplementary Figures S4-S7 show the seasonal surface maps of 500 m and 1000 m for summer (Fig. S4), autumn (Fig. S5),

25   winter (Fig. S6) and spring (Fig. S7). All subsequent figures' numbers in the main manuscript and in the supplementary have been changed accordingly. Given the key oceanographic importance of the representation of the deeper regions in the NAPv1.0, we left much of the original text. However, we have changed the order of presentation to make the text clear and hopefully easier to read. See Lines 249-284 of the revised manuscript.

30